

# Lava flow hazard at Fogo Volcano, Cape Verde, before and after the 2014-2015 eruption

Nicole Richter[1], Massimiliano Favalli[2], Elske de Zeeuw-van Dalfsen[1], Alessandro Fornaciai[2,3], Rui Manuel da Silva Fernandes[4], Nemesio Perez Rodriguez[5], Judith Levy[1], Sónia Silva Victória[6] and
Thomas R. Walter[1]

[1] German Research Centre for Geosciences (GFZ), Potsdam, 14473, Germany
[2] Istituto Nazionale di Geofisica e Vulcanologia (INGV), Pisa, 56126, Italia
[3] Dipartimento di Fisica e Astronomia (DIFA), Alma Mater Studiorum - Università di Bologna, Bologna, 40127, Italia
[4] Instituto D. Luiz, University of Beira Interior, Covilhã, 6201-001, Portugal
[5] Instituto Tecnológico y de Energías Renovables (ITER), Granadilla de Abona, 38611, España
[6] Universidade de Cabo Verde, Praia, Cabo Verde

*Correspondence to:* Nicole Richter (nrichter@gfz-potsdam.de)

**Abstract.** Lava flow simulations help to better understand volcanic hazards and may assist emergency preparedness at active volcanoes. We show that at Fogo Volcano, Cape Verde, such simulations can explain the 2014-2015 lava flow crisis and
therefore provide a valuable base to better prepare for the inevitable next eruption. In a rapid disaster response effort, we conducted topographic mapping in the field and a satellite based remote sensing analysis. We produced the first topographic model of the 2014-2015 lava flows from combined Terrestrial Laser Scanner (TLS) and photogrammetric data. This high resolution topographic information facilitates lava flow volume estimates of $43.7 \times 10^6$ m$^3$ (+/- $5.2 \times 10^6$ m$^3$) from the vertical difference between pre- and post-eruptive topographies. Both, the pre-eruptive and updated Digital Elevation Models
(DEMs) serve as the fundamental input parameters for lava flow simulations using the well-established DOWNFLOW algorithm. Based on thousands of simulations, we assess the lava flow hazard before and after the 2014-2015 eruption. We find that, although the lava flow hazard has changed significantly, it remains high at the locations of two villages that were destroyed during this eruption. This result is of particular importance as villagers have already started to rebuild the settlements. We also analyse satellite radar imagery acquired by the German TerraSAR-X (TSX) satellite to map lava flow
emplacement over time. We obtain the lava flow boundaries every 6 days during the eruption which assists the interpretation and evaluation of the lava flow model performance. Based on this, we discuss how our study can help improving the general understanding of basaltic lava flow behavior. Our results highlight the fact that lava flow hazards change as a result of modifications of the local topography due to lava flow emplacement, which implies the need for up-to-date topographic information in order to assess lava flow hazards. We also emphasize that areas that were once overrun by lava flows are not
necessarily "safer", even if local lava flow thicknesses exceed the average lava flow thickness. Our observations will be important for the next eruption of Fogo Volcano and have implications for future lava flow crises and disaster response efforts at basaltic volcanoes elsewhere in the world.

## 1 Introduction

Effusive volcanic eruptions are associated with lava flows that may cause damage and long-lasting impact on infrastructure
and economy. In Italy, the village San Sebastiao was destroyed by Mount Vesuvius' lava flows in 1944 for the third time in less than 100 years and yet was rebuilt again (Kilburn, 2015). In January 2002, lava flows advancing from Nyiragongo Volcano overran the city of Goma in the Democratic Republic of Congo, which was also rebuilt on top of the 2002 lava flow (Chirico et al. 2009). Destructive effusive volcanic eruptions also occur frequently at places such as the Island of Hawai'i (Kauahikaua and Tilling, 2014; Poland et al., 2016), or at Mount Etna, Sicily, Italy (Favalli et al, 2009b). Yet, a common
observation in many of these classic examples is that, for various reasons, residents rebuild their houses and return to live in





hazardous areas. Studying effusive eruptions and understanding the mechanisms of lava flow emplacement over time, as well as lava flow hazard assessment and the proposal of mitigation strategies therefore have developed into fundamental branches of volcano sciences. Recent crises, such as the 2014-2015 Pahoa lava flow crisis at Kīlauea Volcano, Hawai'i (Poland et al., 2016), and the highly destructive 2014-2015 eruption of Fogo Volcano, Cape Verde (González et al., 2015),

have again shown that up-to-date lava flow hazard information is needed in inhabited volcanic environments and that this information has to be effectively communicated to the officials in charge of public safety and to the residents of areas at risk. Fogo Island features a prominent giant landslide amphitheater that has been gradually filled by eruptive materials since the flank failure in the Quaternary period and continues to be filled during recent eruptions. The flat plain of the Chã das Caldeiras (Chã, cf. Fig. 1), the Pico do Fogo stratocone, and the eastern flank of the post-collapse sequence are parts of this

infilling progression (Day et al., 1999). In the 1860s the first settlements were established within the Chã, likely because of the abundance of fertile volcanic soils. Since that time three major eruptions occurred at Fogo Volcano (1951, 1995, and 2014-2015), all of which affected the Chã. After the 1995 Fogo eruption, lava flow simulations were tested on the base of a 15 m Digital Terrain Model (DTM) (Quental et al., 2003). The Cellular Automata (CA) technique was applied to simulate the time and space dependent flow emplacement. Results were in agreement with the actual lava flow coverage, but only the

first two days of the 1995 eruption were reproduced successfully. A general, yet provisional, volcanic hazard map for the scenario of a renewed phreatomagmatic explosive eruption comparable to the major 1680 eruption was provided by Jenkins et al. (2014) on the base of investigations by Day and Faria (2009, unpublished). This map suggested that the entire area of the Chã as well as the eastern flank of the volcano were areas of high hazard resulting from lava flows, 2-10 m ash fall, possible pyroclastic surges and rock avalanches. A more thorough lava flow hazard assessment for effusive scenarios at

Fogo Volcano on the base of lava flow simulations was lacking and needed to be addressed. Therefore, one major aim of this study is to construct probabilistic lava flow hazard maps for Fogo Volcano. We are particularly interested in whether the 2014-2015 eruption has significantly changed the lava flow hazard in the affected areas, as this has important implications in temporal and spatial changes of lava flow hazards in general.

A variety of algorithms have been developed with the common aim of understanding the dynamics of lava flow

emplacement, forecasting lava flow paths, and constructing lava flow hazard maps (e.g. Favalli et al., 2005; Harris and Rowland, 2001; Del Negro et al., 2008). These algorithms have been applied to numerous volcanoes, including but not limited to Nyiragongo Volcano, Mount Cameroon, and Mount Etna (Favalli et al., 2009a, 2011b; Tarquini and Favalli, 2011). Modelling techniques follow either the probabilistic or the deterministic approach. The MAGFLOW simulation code (Del Negro et al., 2008) is a deterministic approach that relies on pre-existing knowledge or at least simplified assumptions

about the physical and rheological characteristics of flowing lava (Cappello et al., 2015; Tarquini and Favalli, 2015). The FLOWGO model by Harris and Rowland (2001), also a deterministic model, allows for the simulation of changing physical properties, e.g. changes in velocity or thermorheology, of flowing basaltic lava following a predefined channel downslope (Harris et al., 2015). Here we use the DOWNFLOW probabilistic code (Favalli et al., 2005) to create lava flow hazard maps. Based on the gravitational law, DOWNFLOW follows the simple assumption that lava flows downhill from an eruption site.

DOWNFLOW is known to work well at steep terrain (Favalli et al., 2009a, 2011b; Tarquini and Favalli, 2011). Here we apply the model to a rather flat area, the Chã das Caldeiras. One main advantage of this code over deterministic models is that only basic physics apply and that therefore no pre-existing knowledge or assumptions on physical properties of the lava flows are needed. The most important input for the DOWNFLOW simulation is an accurate and up-to-date Digital Elevation Model (DEM) and the location of the eruptive vent.

High resolution topographic information does not only serve as an essential pre-requisite for lava flow simulations, it is also one of the first needs upon any effusive eruption as it allows for lava flow thickness and volume estimates. Modern remote sensing techniques, such as photogrammetry, airborne and terrestrial Light Detection And Ranging (LiDAR), terrestrial laser scanning, and Synthetic Aperture Radar Interferometry (InSAR) are among the most commonly used sources of terrain



information for detailed analyses of the Earth's surface. Major differences between these methods relate to the achievable spatial resolution and coverage, as well as information quality and accuracy. The decision of which method to use highly depends on the specific application and the user's needs. Modern high resolution satellite systems, such as Pleiades (optical) and TanDEM-X (radar), need to be tasked to acquire topographic data in response to a volcanic crisis. Immediate and

effective updates of pre-existing topographic information can be achieved in a more flexible manner using ground-based technologies, such as terrestrial laser scanning and camera- or drone-based photogrammetry. This especially applies to effusive volcanic eruptions, where only the directly affected areas need to be updated. The potential of very-long-range TLS instruments to survey the dynamics of active lava flow fields and to map the topographic changes associated with the emplacement of new flows was shown at Mount Etna, Italy (James et al., 2009). We produced the first post-eruptive

topographic map of the 2014-2015 Fogo lava flow using TLS and ground-based photogrammetric data in order to update a pre-eruptive photogrammetric DEM of Fogo Island, featuring a 5 m spatial resolution. We estimated lava flow characteristics, such as lava flow thickness and volume. We also generated and compared pre- and post-eruptive lava flow hazard maps. For the first time a TLS DEM was used to update a DEM of a rather large, caldera-like area in rapid response to a volcanic eruption. The use of TLS data as a base for the construction of probabilistic lava flow hazard maps is also new.

In the first section of this paper we describe the geologic setting of Fogo Island and the evolution of the 2014-2015 eruption. We then focus on the four main aspects of our study: 1. Satellite observations of lava flow emplacement, 2. Topographic data acquisition and analysis, 3. Simulation of the 2014-2015 lava flow, and 4. Pre-and post-eruptive hazard assessment. We first describe all the methods used in this study, before presenting the results for each of the four aspects mentioned above. Finally, we discuss the limitations and implications of each individual aspect. We provide a general conclusion at the end of

this publication and make the generated maps available in kml-format in the supplementary material.

**2 Geologic setting and eruptive history**

Fogo Island is one of the youngest volcanic islands of the Cape Verde Archipelago in the Atlantic Ocean and is built up from the remnants of one single giant volcano, known as the Monte Amarelo Volcano. The eastern coastline reflects a catastrophic flank collapse event in the island's geologic history (Day et al., 1999; Ramalho et al., 2015), which is thought to date back

~73 ka (Ramalho et al., 2015). This event left a prominent east facing, extremely steep collapse structure, the Monte Amarelo escarpment or Bordeira. It reaches an elevation of up to 1,000 m above the relatively flat, 9 km wide, NS elongated, caldera-like plain, called the Chã das Caldeiras (Chã). The Chã is located at an average elevation of ~1700 m and covers an area of ~35 km$^2$ (Fig. 1). To the east the Chã is bound by the highest point of the island, the Pico do Fogo stratocone (2829 m). The WSW flank of Pico do Fogo was active from a smaller cone, Pico Pequeno, during both, the 1995 and 2014-

2015 eruptions (Fig. 1).

Reports of frequent eruptive activity at Fogo Volcano exist for about 160 years upon the time of Portuguese discovery and settlement in ~1500 AD (e.g. GVP, 2014). Since 1761 eleven confirmed eruptions occurred (GVP, 2013). The 1785, 1799, 1816, 1847, 1852 and 1857 eruptions produced lava flows travelling seawards on the eastern flank of the volcano. However, the last three eruptions (which occurred in 1951, 1995 and 2014-2015) took place within the Chã, where villages and

agricultural lands are at risk (Torres et al., 1997 in Texier-Teixeira et al., 2014).

On 2 April 1995 a fissure eruption started at Pico Pequeno (Amelung and Day, 2002), a small cone WSW of Pico do Fogo (Fig. 1). All residents were evacuated, but houses as well as ~4.3 km$^2$ of agricultural land were destroyed (GVP, 1995a). The total area covered by lava flows during this eruption was estimated to be around 4.7 km$^2$ (Amelung and Day, 2002). The flow thickness ranged between 1 m and ~20 m (GVP, 1995b; Worsley, 2015). After a period of increased strombolian

activity, the eruption ended on 26 May 1995. Because some of the best farmland on the island is located within the Chã, people moved back after the eruption despite attempts from officials to relocate the population.



Almost 20 years later, on 23 November 2014, a new eruption started at Pico Pequeno with the opening of a fissure located ~200 m southeast and roughly in parallel to the 1995 fissure (González et al., 2015). At first, six active vents were emitting lava fountains, ash and gas. Later on activity focused at one major vent (located at 24.35341° W and 14.9446° N). Lava flows were emitted from the base of Pico Pequeno and travelled to the southwest before splitting into two main lobes that we will refer to as the northwest (NW) and south (S) lava lobes throughout this paper. The main road and evacuation route out of the Chã was already cut off two days after the eruption had started and the residents of the Chã were forced to transport their movable property uphill before being evacuated from the area (Fig. 2a). The NW lava flow continued to engulf Portela and Bangaeira in early December 2014 and then gradually ceased (Fig. 2b). However, a new lava lobe split from the NW flow closer to the eruption site and advanced westward towards the Bordeira wall (referred to as W lobe throughout this paper). This lava lobe split into a northern and a southern lobe close to the Bordeira wall and covered the houses of the small agricultural settlement of Ilhéu de Losna. In early 2015, effusive activity was replaced by increased strombolian explosions at the vent (cf. Appendix A) before the eruption ended on 8 February 2015 (GVP, 2014). Shortly after the end of the 2014-2015 eruption, reconstruction of buildings and infrastructure within the Chã had started again (Fig. 2c), despite reenacted efforts from officials to suppress permanent residence within the Chã.

## 3 Data and Methods

One of the first needs immediately upon an effusive crisis is an up-to-date model of the new topography. In rapid response to the 2014-2015 Fogo eruption, a GFZ Hazard and Risk Team (HART) processed high resolution satellite radar data and went to Fogo Island in order to acquire high resolution topographic data between 11 January 2015 and 21 January 2015. The topographic data are needed to estimate lava flow characteristics, such as erupted volumes, and serves as the most crucial input data for our lava flow simulations and hazard assessment. The frequent Synthetic Aperture Radar (SAR) satellite data acquisitions allow us to map lava flow emplacement over time, information that help us to better understand the lava flow model performance.

### 3.1 SAR data

We used SAR data acquired by the German satellite TerraSAR-X (TSX) to monitor the emplacement of the 2014-2015 lava flow. The TSX satellite operates at a wavelength of 3.1 cm (X-band). The data were acquired in the satellite's Spotlight mode (~1 m spatial resolution, scene size ~10 km x ~10 km), horizontal polarization, and have a repeat time of 11 days. TSX data are acquired over Fogo Island on four tracks, two ascending (orbital paths 57, incidence angle 53.5° and 148, incidence angle 38.9°) and two descending (orbital paths 64, incidence angle 27.9° and 155, incidence angle 46.3°). The temporal offset between the tracks ranges between ~12 h (for the ascending and descending pairs 57/64 and 148/155) to ~6 days. We take advantage of all available satellite data to monitor lava flow emplacement at least every 6 days between 14 November 2014 and 26 January 2015 using coherence measurements.

Interferometric coherence is a measure of the correlation between the phase components of two SAR images of the same track (i.e. the same viewing geometry) (Hanssen, 2001). Coherence values range from 0 (low coherence, decorrelation) to 1 (high coherence, strong correlation between SAR acquisitions). As a consequence of the time delay between two acquisitions (11 days for TSX), temporal decorrelation occurs in repeat-pass InSAR as the scatterers within a resolution cell move, change their dielectric properties or are replaced by a new set of scatterers (e.g. upon lava flow emplacement or ash deposition) (e.g. Zebker et al., 1996).

### 3.2 Topographic data

We created an updated DEM for Fogo Volcano, which is composed of three different datasets: firstly, a commercial pre-eruptive DEM acquired by GRAFCAN (Sect. 3.2.1), secondly, our TLS DEM that is composed of eight combined point



clouds that cover in total 87.7 % of the 2014-2015 lava flow (Sect. 3.2.2), and finally, four separate, very small DEMs produced by applying the Structure from Motion (SfM) method to optical camera data (Sect. 3.2.3) in order to fill the remaining gaps. We merge the datasets by minimizing the vertical distance between our TLS and SfM point clouds and the commercial pre-eruptive reference DEM using the Minuit minimization tool (Sect. 3.2.4).

### 3.2.1 Pre-eruptive data

We use a commercial pre-eruptive DEM of Fogo Island featuring a 5 m pixel spacing that was generated from contours on the base of photogrammetric data. The DEM was georeferenced using a limited number of points acquired during a mapping campaign in 2003-2004. According to GRAFCAN, the company who generated the DEM, the horizontal and vertical accuracies are 40 cm and 50 cm, respectively. However, the data were delivered in integer values. We therefore expect the accuracy to be smaller. From the delivered DEM grid we generated contour lines with 1 m spacing (in elevation) and re-interpolated the surface.

### 3.2.2 Post-eruptive TLS data

We used a TLS instrument of the type Riegl VZ-6000, with the capability to produce sub-meter resolution topographic data of target objects at a maximum distance of 6000 m from the scanner position. The instrument combines the 3D laser scanning and laser ranging techniques. Data are acquired by the controlled deflection of a laser beam into different directions by means of an oscillating mirror, and the 360° rotation of the scanner's head (3D laser scanning). The distances for each point in the point cloud is determined by measuring the time delay between the emission of the laser pulses and the reception of the target echoes (laser ranging). The coordinates of each single point are collected relative to the origin, along with additional attributes, e.g. reflectance and amplitude. The raw point cloud is stored in a local, scanner-centered coordinate system (Riegl, 2013).

During the field campaign in January 2015, we acquired eight TLS point clouds, which were used for the generation of the updated topographic map (Table 1). We chose three main scanner locations, namely Monte Beco, Monte Saia and Monte Amarelo (Fig. 3). The terrestrial laser scanning technique is commonly associated with the occurrence of shadow areas due to the acquisition geometry. We minimize the shadow effects by scanning from slightly different positions at Monte Beco and Monte Amarelo so that our TLS data points cover 87.7 % of the area overrun by the 2014-2015 lava flow and most of the Chã. We combined the individual scans for these two main scan locations separately by means of a minimum of three common tie points (reflectors) that we installed in the field. Rough orientation of the point clouds was done using handheld GPS positions of the reflectors. At five of the scanner positions, we acquired differential GPS data (Table 1). At each site, GPS data were collected for at least 1h (up to 3h30m) and at one site (SAIA1) data were acquired on two different days. Differential GPS processing was done in two main steps. At first, the average positions of a network of seven permanently installed GPS stations were estimated with respect to the ITRF2008 global reference frame (Altamimi et al., 2011). The seven GPS stations were installed and are maintained by C4G (Co-laboratory for Geosciences, Portugal) and INMG (Instituto Nacional de Meteorologia e Geofísica, Cape Verde) and continuously acquired GPS data during the 2014-2015 Fogo eruption (Fernandes et al., 2015). The computation of the reference positions was done using the Precise Point Positioning strategy (PPP) implemented in the GIPSY-OASIS software package (Zumberge et al., 1997). More details on the here used methodology are provided by Neves et al. (2014). Secondly, the coordinates of the TLS stations were estimated with respect to the permanent network using the TBC (Trimble Business Center) software. This software allows for the adjustment of the position of the TLS sites by estimating the baselines between all simultaneously observed points. The scanner locations and the coverage of the acquired point clouds are shown in Fig. 3a. Details on the scanner positions, and tie point registration accuracies are listed in Table 1.





### 3.2.3 Post-eruptive photogrammetric data

Photogrammetric data were collected from positions along the upper Bordeira ridge using dSLR 15.1-megapixel Canon EOS Rebel cameras with CMOS sensor. We used the SfM method as introduced by Verhoeven (2011) to generate four small DEMs from a total of 77 camera images. A first, rough georeferentiation was performed using manually selected Ground

Control Points (GCPs) before following the error minimization procedure described below. This way we were able to fill 92 % of the remaining TLS data gaps (red areas in Fig. 3b).

### 3.2.4 Data merging

In order to merge the separate point clouds a common reference frame is needed. We therefore minimize the Root Mean Square Error (RMSE) between the TLS and SfM point clouds and a reference DEM using the Minuit2 5.18/00 package,

developed at CERN (James and Winkler, 2004 and references therein). Minuit is a tool to find the minimum value of multi-parameter functions and can be freely downloaded (http://www.cern.ch/minuit). Throughout this paper we will refer to RMSE to actually address the root mean square residuals (in elevation) between our data and the reference DEM, rather than a true absolute error of our data. We use the pre-eruptive DEM as a reference for our point clouds. For the minimization procedure, the Bordeira wall (which is well covered by the TLS point clouds, but mainly interpolated within the pre-eruptive

DEM), as well as the area covered by the 2014-2015 lava flow, are masked out from the pre-eruptive DEM. Therefore, the number of points that were used for the Minuit minimization is smaller than the total number of points in the point clouds (cf. Table 1, columns "# Points (used)" vs. "# Points (filtered, cleaned)"). For those scans, where accurate GPS locations were available, we fix the center of rotation and translation of the cloud at the scanner position. Otherwise all rotation parameters, i.e. rotation around x (roll), y (pitch), z (yaw), and the center of rotation and translation were kept free to adjust.

Iterative runs of the Minuit minimization function are used to decrease the RMSEs. The final minimum RMSEs for the individual and combined point clouds are listed in Table 1. We generate a DEM from more than 164 million TLS data points and mosaic it with the pre-eruptive DEM. This initial post-eruptive DEM was used as a reference DEM for the SfM point clouds (using masks for the areas that were not covered by the TLS data). The achieved Minuit accuracies for the four individual SfM DEMs are listed in Table 2. The masks that were used to estimate the Minuit RMSEs are illustrated in

Appendix B. In a last step the initial post-eruptive DEM was updated using the SfM DEM patches.

### 3.3 Lava flow simulation model

In order to assess the lava flow hazard at Fogo Volcano, Cape Verde, we used the probabilistic lava flow simulation code DOWNFLOW (Favalli et al., 2005). A lava flow simulation follows a number of $N$ steepest descent paths (runs) originating from a known vent location. At each pixel of the DEM grid that is inundated by a lava flow, and for each run, a random

perturbation within a given interval $+/-\Delta h$ is introduced in order to allow widening, branching, and overcoming small obstacles. The parameter $\Delta h$ is the critical parameter that needs to be assessed for each individual volcano. Multiple test runs and a tuning process were performed to find the best fit between the model runs and the coverage of the real lava flow of Fogo Volcano (see Sect. 3.3.2 and Fig. 4). Along the steepest descent paths, topographic lows are filled by the algorithm. The grid points that are crossed more often by maximum slope paths are more likely to be invaded by a lava flow. More

details on the DOWNFLOW simulation are described by, e.g., Favalli et al. (2005; 2011a) and Tarquini and Favalli (2015).

### 3.3.1 Simulation of the 2014-2015 lava flow

The coordinates 24.35341° W and 14.9446° N at an elevation of 1995 m a.s.l. were chosen as source location of the lava flow that erupted on 23 November 2014, based on our high-resolution post-eruptive TLS DEM. It is important to mention that we did not apply a lava flow length constraint at this point, meaning that we did not stop the lava flow simulation when

it reached a certain length.





### 3.3.2 Model calibration

For calibration we analyzed the whole parameter space in order to maximize the fit, $\mu$, between the real lava flow and the DOWNFLOW simulation. As we do not implement a length constraint at this point, the simulation is cut to the length of the real lava flow. The parameters to be calibrated therefore are $\Delta h$ and $N$. The best fit, $\mu$, is calculated according to the
following equation:

$$\mu = \frac{A_S \cap A_R}{A_S \cup A_R}, \qquad (1)$$

where $A_S$ is the area covered by the simulation and $A_R$ is the area covered by the real lava flow. Figure 4 shows the DOWNFLOW calibration for the 2014-2015 Fogo eruption. According to the graph, the value of $N$ is not critical for the fit $\mu$. We choose the value $N = 10,000$ because this number is a compromise between low computational time and statistical
robustness (Favalli et al., 2009a). The maximum fitness of $\mu = 0.47$ is achieved using a parameter $\Delta h = 3$ m.

### 3.4 Hazard map generation

Using a single DOWNFLOW simulation we reconstructed the 2014-2015 lava flow which originated from one known vent (Sect. 3.3.1). The simulation of a future scenario requires a fundamental lava flow hazard assessment; in our case this includes the simulation of lava flows for ~82,000 vents. In order to create meaningful hazard maps, these simulations need to
be weighted by two factors: (a) the lava flow length constraint, in order to stop the flows before they reach the end of the dataset (Sect. 3.4.1) and (b) the probability function of future vent opening, because in reality a vent opening at some locations is more likely than at others (Sect. 3.4.2).

### 3.4.1 Lava flow length constraint

For the lava flow length constraint, information on historic eruptive vents and corresponding lava flow lengths is needed. For
other volcanoes it was shown that a negative correlation exists between vent elevation and lava flow lengths, e.g. at Mount Etna (Favalli et al., 2009b), Nyiragongo Volcano (Favalli et al., 2009a), and Mount Cameroon (Favalli et al., 2011b). According to these examples, vents located at higher elevations tend to produce shorter lava flows. For Fogo Volcano, we use a geologic map as a base of information (Torres et al., 1997 in Texier-Teixeira et al., 2014). We also take into account an updated map published recently by Carracedo et al. (2015). We georeferenced the maps using ~100 manually defined control
points, determine the historic vent locations and plot the vent elevations against the lengths of the corresponding lava flows (Fig. 5). This plot distinguishes lava flows that reach the sea from lava flows that stop onshore. While the first only help to find the minimum lava flow length, the latter are used as the base for the length constraint. A Fogo-specific limitation is that we only have a record of five historic lava flows that did not reach the sea, and all of the corresponding vents are located within a narrow elevation range of less than 300 m (between 1,760 m and 2,020 m a.s.l.). Therefore, we cannot infer a
negative correlation between vent elevation and corresponding lava flow lengths. Instead, we have chosen to apply a lava flow length range for the Fogo-specific vent elevations. We apply a buffer zone to the observed minimum and maximum lava flow lengths given by the average distance between consecutive flow lengths (approximately 1,000 m). We consider a minimum and maximum lava flow lengths constraint of 3,000 m and 9,000 m, respectively.

### 3.4.2 Probability density function of vent opening

We base the probability of vent opening on the record of historic vents assuming that future vents are more likely to open in areas where historic vents cluster. The vent locations we used were provided by our post-eruptive DEM and the geologic map of Fogo Volcano (Torres et al., 1997 in Texier-Teixeira et al., 2014). They were used to estimate the Probability





Density Function (PDF) of vent opening for Fogo Volcano by applying a Gaussian smoothing kernel with a bandwidth of 800 m (Bowman and Azzalini, 2003; Favalli et al., 2011b), which integrates up to be 1 (i.e. 100 %) over the entire domain.

We describe two different scenarios and create the corresponding hazard maps, namely the "Pico Pequeno scenario" and the "Fogo scenario". The first scenario considers a vent opening at Pico Pequeno. We consider a future vent opening at this location to be a likely future scenario, as the two most recent eruptions occurred from Pico Pequeno. For this scenario the entire PDF domain is a circle of 1,150 m in diameter centered between the 1995 and 2014-2015 vents (circle in Fig. 6). We consider a PDF that is constant inside this circle and 0 outside. The second scenario applies to Fogo Volcano as a whole. For the "Fogo scenario" 42 vents within the Chã were considered (Fig. 6), as we do not have any record of historic eruptions outside the Chã. In the "Fogo scenario" (Fig. 6), the area at and around Pico Pequeno (as indicated by the circle) has an overall probability of vent opening of ~2 %.

### 3.4.3 Probability of lava flow invasion

We create lava flow hazard maps for Fogo Volcano that show the probability $H_i$ of any pixel $i$ to be inundated by a future lava flow, before knowing where the vent will be located. This probability is given by the following equation (Favalli et al., 2011b):

$$H_i = \sum_j \rho_{Vj} \Delta x \Delta y \cdot P_{ij} \cdot P_{Lij}, \qquad (2)$$

where the sum extents over all possible vent locations $j$ with the coordinates $x_j$ and $y_j$, $\Delta x$ and $\Delta y$ are the dimensions of the resolution cell, $\rho_{Vj}$ represents the PDF of vent opening (cf. Sect. 3.4.2 and Fig. 6) at vent $j$. $P_{ij}$ is a mask that will have the value 1, if a DOWNFLOW simulation starting from the vent $j$ covers the pixel $i$, otherwise $P_{ij}$ will be 0. Finally, $P_{Lij}$ represents the lava flow length constraint (cf. Sect. 3.4.1 and Fig. 5) and gives the probability that a DOWNFLOW simulation will reach pixel $i$ when originating at vent $j$. Following Fig. 5, $P_{Lij} = 1$ at a length of ≤3000 m, $P_{Lij} = 0$ at a length of ≥9,000 m, and $P_{Lij}$ is interpolated between 3,000 m and 9,000 m. For the "Fogo scenario", we consider 82,000 future vents, i.e. one vent every 50 m, distributed on a square mesh. For the "Pico Pequeno scenario" 413 vents were taken into account within the given circle as indicated in Fig. 6.

In practice this means we perform 82,000 DOWNFLOW simulations, each with $N = 10,000$ and $\Delta h = 3$ m, and store them in a simulation database. For a given simulation database (i.e. a defined, fixed $P_{ij}$), all other parameters of the equation above are easily adjustable. Consequently, we can investigate different scenarios by changing the lava flow lengths constraint or the vent opening PDF. The hazard maps for the "Pico Pequeno" and "Fogo" scenarios, for example, are created using two different vent opening PDFs.

Furthermore, lava flow hazard maps are created for two time steps based on the two available DEMs; the pre-existing, pre-eruptive DEM is used to reconstruct the 2014-2015 lava flow and to assess the pre-eruptive lava flow hazard within the Chã and on the eastern flanks of the volcano. We then repeat the same simulations using the updated DEM. We assess to which extent the most recent eruption has changed the probability of lava flow inundation in the affected area and the whole eastern part of the island.

## 4 Results

### 4.1 Lava flow mapping from satellite radar data

Figure 7a shows the coherence between two SAR acquisitions acquired on 14 November 2014, nine days prior to the start of the eruption, and on 25 November 2014, two days after the start of the eruption. The areas shown in blue color have a low coherence because they were covered by lava flows within the first two days of the eruption. Other blue areas close to the vent and on the western flank of the Pico do Fogo stratocone are incoherent due to vent opening and ash deposition. This



coherence map shows that the NW lava lobe had already traveled almost 4 km within the first two days of the eruption, but had not quite reached the village of Portela yet. The S lava flow was also already emplaced (Fig. 7a). This lava lobe did not advance much after emplacement, except for some minor widening (Fig. 7a – 7h). Between 20 November 2014 and 1 December 2014, primarily the NW lava flow widened, destroying the first houses of Portela. We also observe minor

propagation and widening at the W lava lobe (Fig. 7b). This trend of widening and engulfing more houses of Portela continued throughout 6 December 2014 (Fig. 7c). Up to that time, lava flowed in a well-defined channel north of Monte Saia (Fig. 7a – 7c, cf. Fig. 8a, point #1 and Fig. 8b, profile C-C'). Most of Portela and Bangaeira, according to our coherence maps, were covered by lava flows in the period between 6 December 2014 to 12 December 2014 (Fig. 7c and 7d). Until 17 December 2014 the W lava lobe was propagating, but the settlement of Ilhéu de Losna was not yet harmed (Fig. 7a – 7e).

Between 17 December 2014 and 23 December 2014, this lava flow advanced further towards the west, where it split into two N and S sub-lobes after reaching the Bordeira wall (Fig. 7e and 7f). At this point the third, smaller settlement of Ilhéu de Losna was destroyed. Since that time until the end of the eruption, the lava flow extent stabilized, implying that effusive activity had slowed down. Only minor widening was observed at the W lava lobe (Fig. 7f and 7g). Also, during fieldwork, we observed an active surface flow on 12 January 2015 and minor propagation of the lava flow in-between Monte Saia and

Monte Beco (Fig. 7g). During that time, cooling caused areas of incoherence on the fresh lava flows and explosive activity produced incoherence at the vent (Fig. 7g). Details on the active surface flow are provided in Appendix C. Even though the eruption lasted until 8 February 2015, no lava flows were active in the last period of the eruption. The final boundary of the 2014-2015 lava flow as shown in Fig. 7h (and as a dashed line in Fig. 7g) encloses an area of 4.85 km².

## 4.2 Topographic model

We achieve a RMSE of 1.08 m for the global geolocation of our final post-eruptive point cloud (cf. Table 1). This value was calculated using the Minuit minimization procedure between the final, combined point cloud and the pre-existing DEM. The error is smaller when comparing post-eruptive and pre-eruptive grids (RMSE = 0.89 m). Moreover, when estimating this error only within the most representative area around the lava flow, as shown by the masks in Appendix B, the RMSE is further reduced to 0.81 m. However, after filling the remaining gaps, the overall RMSE between the filled DEM and the pre-

existing DEM is 1.08 m. With this accuracy, we provide a post-eruptive DEM featuring a 5 m pixel resolution, which we consider updated to 16 January 2015.

The difference between the pre- and post-eruptive DEMs is a measure for lava flow characteristics, such as the total area covered by lava, the lava thickness (Fig. 8b) and the total lava volume. The 2014-2015 lava flow coverage is 4.84 km$^2$ featuring an average thickness of 8.9 m, calculated from the vertical DEM difference. The total erupted lava volume, updated

to 16 January 2015, is estimated to be 43.2 x 10$^6$ m$^3$. However, from the RMSE between the pre- and post-eruptive DEMs, a maximum error in volume is calculated to be 5.2 x 10$^6$ m$^3$, corresponding to 12 % of the total lava flow volume.

Two additional error sources may lead to an underestimation of the total lava flow volume in our study. Firstly, data gaps still exist for an area of 46,579 m², or 1 % of the area covered by the lava flow (dark red areas in Fig. 8b), introducing an error of about 1 % to our lava flow volume estimate (~0.43 x 10$^6$ m$^3$). Secondly, we do not include any surface flows that

were active after 16 January 2015 (the acquisition date of scans MBC 4 and MBC 5, cf. Table 1). However, we observed an active lava flow in the morning of 12 January 2015 that was still growing after 17 January 2015, according to our multi-temporal TLS analysis described in Appendix C. We estimate the volume of the growing lava flow that is not covered by our data to roughly amount to ~0.05 x 10$^6$ m$^3$ (Appendix C). Therefore, our best total erupted volume estimate for the 2014-2015 flank eruption of Fogo Volcano is 43.7 x 10$^6$ m$^3$ +/- 5.2 x 10$^6$ m$^3$.

According to Fig. 8b, the 2014-2015 lava flows have a maximum thickness of ~35 m close to the vent. The lava flows are thicker where topography was filled by lava ponding. One very significant ponding area is located west of Portela (Fig. 8b,





profiles A-A' and B-A'). Here, lava flows have a maximum thickness of ~25 m, while at the location of the villages, lava flows are maximal 8 m - 9 m thick.

### 4.3 Simulation of the 2014-2015 lava flow

Using the DOWNFLOW simulation, we reconstruct the 2014-2015 lava flow (Fig. 8a). Overall, 75 % of the actual lava flow
area is covered by the DOWNFLOW simulation. We find intriguing similarities between the simulation and the lava flow thickness in several areas, including the initial flow (through point #1 in Fig. 8a), as well as the NW (#2), W (#3), and S (#4) lava lobes. We observe that single flows, i.e. flows that are not affected by multiple phases of emplacement (e.g. #1, #5, and #4), are reproduced well by the DOWNFLOW simulation. However, lava flows that are emplaced in later effusive pulses are not as well represented (points #6 and #7). Pixels located within topographic ponds are hit by a simulation more often, due to
the implemented filling algorithm of the DOWNFLOW code.

### 4.4 Lava flow hazard assessment

Our hazard maps show the probability of lava flow invasion for both, pre- and a post-eruptive topography, and the "Fogo" -, and "Pico Pequeno" scenarios.

#### 4.4.1 Lava flow hazard before the 2014-2015 eruption

This period we refer to as "pre-eruptive". The pre-eruptive hazard maps are calculated on the base of the pre-eruptive DEM and show the lava flow hazard after the end of the 1995 eruption, but before the onset of the 2014-2015 eruption (on 23 November 2014).
Figure 9a shows the pre-eruptive hazard map for the "Pico Pequeno scenario". We find that the locations of the villages (#1, #2, and #3 in Fig. 9a) are zones of very high lava flow hazard (more than 75 %). By comparing the hazard map to the outline
of the actual lava flow we find that any vent opening within the given circle around Pico Pequeno would have resulted in approximately the same lava flow coverage as the 2014-2015 eruption did. In fact, before the eruption, a small portion of the area that is now covered by lava flows had even 100 % probability to be invaded by a lava flow. On the other hand, areas that had 0 % probability of invasion are now covered with lava flows (Fig. 9a, cf. points 6 and 7 in Fig 8a). At the same time, some parts of the area that is surrounded by the NW and W lava lobes (which is actually the area that is covered by the 1995
lava flow (Carracedo et al., 2015)) were almost certain to be invaded according to this map, but were not affected by the 2014-2015 lava flows (Fig. 9a).
The pre-eruptive lava flow hazard map for the "Fogo scenario" is shown in Fig. 10a. Generally areas of high lava flow hazard cluster along the Bordeira wall, but low probability of lava flow invasion can be observed at the Pico do Fogo stratocone. Some of the larger cones within the Chã appear to serve as a barrier and produce a "lava flow hazard shadow area" behind them. This is especially true for Monte Beco, but also for the southern vent of the 1951 eruption and at the 1852
area" behind them. This is especially true for Monte Beco, but also for the southern vent of the 1951 eruption and at the 1852 vent (Fig. 10a, for vent locations cf. Fig. 6). In this map, the area covered by the 1995 flow shows a probability of invasion of more than 7 % close to the borders of the 2014-2015 lava flow. As for the volcanoes' eastern flank, comparably high lava flow hazard exists especially along the edges of the landslide amphitheatre. Considering the fact that this map is not specific for vents around Pico Pequeno, it is striking that all three flow fronts of the 2014-2015 lava flow cover elevated to very high
hazard areas (10 % - 28.4%). For the "Fogo scenario" the two villages of Portella and Bangaeira as well as the small settlement of Ilhéu de Losna were located in these high hazard zones of the NW and W lava lobes.

#### 4.4.2 Lava flow hazard after the 2014-2015 eruption

This period we refer to as "post-eruptive". The post-eruptive hazard maps reveal the probability of lava flow invasion for the next eruption of Fogo Volcano.





The post-eruptive hazard map for the "Pico Pequeno scenario" shows that the lava flow hazard is generally higher in the northern part of the Chã (Fig. 9b), with a maximum probability of lava flow invasion of 88.42 %. Especially closer to the vent (Pico Pequeno), channeling is not as strongly pronounced as before the 2014-2015 eruption (cf. Fig. 9a). The probability of lava flow invasion has not noticeably decreased for the locations of the villages of Portela (point #2 in Fig. 9b)

and Bangaeira (point #1 in Fig. 9b), in fact, for the latter the lava flow hazard has even slightly increased. Only for the small settlement of Ilhéu de Losna (point #3 in Fig. 9b), the probability of lava flow invasion has significantly decreased by 46 %. "Islands" of no lava flow hazard (probability of invasion = 0) within the Chã exist on and behind Monte Beco, in an area west of Pico Pequeno, and also in areas southeast, northeast and directly north of Pico Pequeno. According to the map, lava flows, if they were long enough, would flow to the north of the Chã from where they would continue to flow down the

volcanoes' eastern flank.

The post-eruptive lava flow hazard map for the "Fogo scenario" is shown in Fig. 10b. According to this map, the probability of lava flow invasion during the next eruption of Fogo Volcano lies between 0 % and 29.3 %. Generally, areas along the Bordeira wall are areas of higher overall lava flow hazard. There are regions in the south part of the Chã that were not affected by 2014-2015 flow. There, lava flow hazard zones are distributed very similar as compared to before the 2014-2015

eruption (Fig. 10a and 10b). The locations of the former towns of Portela and Bangaeira are still among the very high hazard zones. The highest probability of invasion exists where lava flows cluster to leave the Chã, overrun the scarp, and flow down the steep north-eastern flank of the volcano. Along this path other villages, like Fonseco, are at risk. At the south-eastern slope, the village of Tinteira is crossed by a strip of a remarkable hazard of ~10 %.

The letter "H" in Fig. 9 and Fig. 10 indicates the location of a new building that was constructed on top of the 2014-2015

lava flow (personal communication Mustafa Kerim Eren, November 2015), at 24.37360° W and 14.96744° N. The lava flow at this point is ~18 m thick. The lava flow hazard for the "Pico Pequeno scenario" at this location was 82.5 % before and is 79.4 % after the 2014-2015 eruption. The "Fogo scenario" hazard maps show a lava flow hazard for the location of the new building of 15.8 % before and 11 % after the eruption.

### 4.4.3 Which future vent locations pose the highest hazard to the villages of Portela and Bangaeira?

To answer this question we provide the area of vent opening that, according to the DOWNFLOW simulation, could potentially result in a lava flow able to invade Portela and Bangaeira (Fig. 11). In this so called "catchment map", areas are ranked according to the minimum length that lava flows need to travel before reaching the village. In other words, it shows in colors the area of future vent openings that potentially threaten Portela and Bangaeira (indicated by polygon). Any vent outside the colored area lays downslope from the villages or belongs to a different catchment and will therefore produce lava

flows that will not harm the area covered by the polygon.

For example, if a future vent opens in the red/orange area, even short lava flows will reach the villages. On the other hand, if a future vent opens in the blue area, the lava flows need to travel a long distance before reaching the villages of Portela and Bangaeira. The length that a lava flow will have to reach before engulfing the villages is given in the number of kilometers in Fig. 11. Technically, such a map can be provided for any pixel or area of interest on the base of our Fogo DOWNFLOW

simulation database.

### 5 Discussion

The 2014-2015 eruption caused a humanitarian crisis on the Island of Fogo. Approximately 90 % of the houses in the villages of Portela and Bangaeira and the small agricultural settlement of Ilhéu de Losna as well as 24.6 % of the farmlands within the Chã were destroyed by lava flows. This left approximately 1,000 people homeless and seriously endangered their




source of livelihood (United Nations, 2015). Because the volcano provides some of the most fertile soils on the island and facilitates geotourism, people are already returning to live in the Chã.

Our ground-based data were acquired in direct response to the effusive crisis at Fogo Volcano. The GFZ Potsdam has over 20 years of experience with such HAzard and Risk Team efforts (HART, formerly "Task Force"). A field team went immediately to Fogo Island to acquire high resolution topographic data of the 2014-2015 lava flows as the lava flow production ceased and collaborated with other research teams, from the Cape Verdes, Spain, Portugal, Italy, and elsewhere. HART efforts are meant to collect data for fundamental research purposes and to generate meaningful scientific results in order to assist local partners and to educate and train residents of hazardous areas. For hazard mitigation to be successful, it is of utmost importance to effectively communicate scientific results as well as the uncertainties related to lava flow hazards to the local emergency management authorities and the general public (Kauahikaua & Tilling, 2014, Poland et al., 2016).

In the following section we will elaborate the limitations and discuss the implications of each of the four main aspects of our study (i.e. satellite data analysis, topographic data analysis, lava flow simulation, and hazard maps).

### 5.1 Lava flow mapping

SAR coherence is known to be a valuable tool to define lava flow boundaries (Zebker, 1996) and continues to be used for lava flow mapping purposes (e.g. Dietterich et al., 2012). Also during the 2014-2015 eruptive crisis of Fogo Volcano, successive flow mapping was done by the Copernicus Emergency Management Service (2014) using SAR coherence information derived from COSMO-SkyMed satellite data. Data sampling was complemented with high resolution optical data from the Pleiades and WorldView-2 satellites. This way a map was produced showing the lava flow boundaries in time steps of 1-7 days in the period between 29 November 2015 and 28 December 2015 (Copernicus Emergency Management Service, 2014). In our study, we generate TSX coherence maps to track lava flow emplacement over time, with the aim to provide an empirical validation for the DOWNFLOW reconstruction of the 2014-2015 lava flow. A common disadvantage of this approach is that decorrelation may occur due to many factors besides lava flow emplacement, factors such as vent opening, vent growing and ash deposition. With our coherence maps we cannot delineate these eruptive processes apart in close proximity to the active vent at Pico Pequeno, causing our lava flow boundaries to be unprecise there. In other volcanic areas, vegetation and snow coverage will further limit the applicability of coherence maps to track flow emplacement (Dietterich et al., 2012), both of which are minimal or nonexistent at Fogo Volcano. As the resulting extent of the mapped (TSX coherence, and vertical DEM difference between the pre- and post-eruptive DEMs) and the simulated (DOWNFLOW) 2014-2015 lava flows match almost perfectly (cf. Fig. 7h, Fig. 8 and Appendix B), we are confident that our data interpretation and model performance are sound. Due to the temporal resolution of the TSX satellite data (11 days) and the time delay between orbital paths, we are able to monitor lava flow emplacement in intervals of about 6 days, sufficient for our application. To study the temporal evolution of the lava flows in even more detail was not anticipated; however, additional SAR and optical data from other satellite systems (e.g. the Sentinel-1 or COSMO-SkyMed missions, or the optical Pleiades and WorldView-2 satellites) provide an even higher data sampling.

### 5.2 Post-eruptive DEM generation and lava flow characteristics

Depending on the instrument, TLS is capable of acquiring topographic data over distances of up to ~6,000 m (Riegl, 2013). However, previous studies have shown that in particular in volcanic areas useful point cloud densities could only be achieved at a maximum distance of ~3500 m (James et al., 2009). Therefore, the technique is usually applied to study smaller areas of ~1-5 km$^2$ in very high spatial (cm-scale) and temporal resolutions (up to 1 scan every 10 minutes) (e.g. James et al., 2009, Jones et al. 2015, Slatcher et al., 2015).

Here we acquired a set of 8 TLS scans to cover ~75 % of the Chã das Caldeiras, which is ~35 km$^2$ in size; another ~1.75 % of the area of the Chã is covered by our photogrammetric data. The main target of our survey was the area affected by the



2014-2015 lava flow, of which we covered 99 % using combined TLS and photogrammetric data. The combination of SfM and TLS data have also been successfully applied in the past (e.g. Pesci et al., 2007). We are lacking data for two small gaps (dark red areas in Fig. 8b) and a small lava flow that was active during the period of our TLS data acquisition. To estimate the resulting error contribution of the active lava flow, we used a multi-temporal TLS analysis (Appendix C).

The acquired TLS point cloud has an unprecedented resolution and quality, sufficient for the presented application, but the most critical remaining limitation of our TLS dataset is the shadowing. In addition to minor shadows resulting from the viewing geometry, we have to mention one major shadow area at the north side of the 2014-2015 vent that had to be interpolated between the pre-existing and the post-eruptive DEMs. Due to the ongoing strombolian explosions during field work, the scanner could not be set-up on the slopes of the Pico do Fogo cone, looking towards the west.

While the TLS data acquisition is time consuming and logistically challenging, its precision is higher and the processing time is faster compared to the SfM technique (Westoby et al., 2012). The general quality of our TLS data is better than that of our photogrammetric data, as camera images were acquired from locations along the upper ridge of the Bordeira wall at a rather large distance of ~3-4 km. Thus, our SfM data are affected by water vapor, the sunlight, as well as unclear skies due to volcanic gas and ash emissions. Therefore, we used TLS data where available and added SfM data only where needed.

The data and techniques used were designed to generate topographic information on the new lava flows; a survey to cover the whole island was not anticipated. For the study at hand we produced a DEM featuring a resolution of 5 m to meet the resolution of a pre-existing dataset. On a more regional scale, where the point cloud is dense, e.g. at the active vent or at the buried villages of Portela and Bangaeira, our data can also be used in a much higher spatial resolution (cm-scale).

From the vertical DEM difference between pre- and post-eruptive DEMs, we estimate the total erupted lava flow volume to
be $43.7 \times 10^6 \, \text{m}^3$ +/- $5.2 \times 10^6 \, \text{m}^3$. We consider this to be the most accurate lava flow volume estimate for the 2014-2015 Fogo eruption at this date. The volume is 4 times larger than previous rough estimations suggested (Ferrucci et al., 2015). According to our results, the 2014-2015 lava flow has a comparable surface coverage ($4.84 \, \text{km}^2$) and volume ($43.7 \times 10^6 \, \text{m}^3$ +/- $5.2 \times 10^6 \, \text{m}^3$) to the lava flow of the 1995 eruption (area: ~$4.7 \, \text{m}^2$, volume: ~$46 \times 10^6 \, \text{m}^3$) (Amelung and Day, 2002).

### 5.3 DOWNFLOW performance

The Fogo case study represents a successful application of the DOWNFLOW algorithm. In this study the DOWNFLOW simulation has proved to perform well on rather flat areas, like the Chã das Caldeiras. This implies that lava flow paths are largely controlled by the topography even, and maybe especially, in relatively flat terrain. Furthermore, our Fogo example demonstrates the first application of DOWNFLOW to a TLS dataset. In order to discuss the DOWNFLOW performance, we compare the simulation (Fig. 8a) to the real lava flow coverage (Fig. 8b and Fig. 7). The very early phase of the 2014-2015
eruption, when lava travelled in a well-defined channel according to the TerraSAR-X coherence analysis (Fig. 7a-c) and the thickness map (Fig. 8b and profile C-C'), is reproduced by the DOWNFLOW simulation in great accuracy (Fig. 8a). Furthermore, it seems that lava flows are thicker where the number of paths crossing a pixel by the simulation, are highest (e.g. #2, #3, #4, and across profile C-C' in Fig. 8b). This applies especially to topographic ponds, where the filling algorithm, which is implemented in the DOWNFLOW model, causes the lava flow simulation to fill in the local topography before
continuing the path downslope (profiles A – A' and B – A'). This also explains the fact that, even though the lava flow had reached the first houses of Portela within the first days of the eruption, it stopped for a couple of days, while ponding, before continuing its path downslope, overflowing the villages of Portela and Bangaeira (Fig. 7, Sect. 2 and Sect. 4.1). We conclude that, both channeling and ponding can be very well simulated by the code.

Differences between the simulation and the real lava flow coverage occur due to two main facts: first, the DOWNFLOW
simulation runs until the lava flows hit the end of the DEM, while the actual lava flows stop when effusive activity ceases. Second, the DOWNFLOW simulation starts only once at the vent location and then keeps running downslope, while the real lava flow is produced iteratively depending on the supply rate at the vent. This process is creating new topography upon





emplacement, which changes the paths of lava emplaced during subsequent effusive pulses. This observation explains why the DOWNFLOW simulation fits the extent of the lava flow during the first few days of the eruption almost perfectly in close proximity to the vent while the distant 2014-2015 lava flow fronts are in good agreement with topographic ponds and therefore a higher number of times that a pixel is hit by a simulation.

With these findings, the 2014-2015 lava flows of Fogo Volcano provide intriguing examples of the impact of local geologic structures, such as topographic channels and ponds, on lava flow pathways and the ability of numerical lava flow simulations to reconstruct and predict these. We suggest that updating the local topography, even during an ongoing eruption, is of importance in order to forecast paths of lava flows produced by subsequent effusive pulses.

**5.4 Lava flow hazard maps**

Our volcano-wide hazard maps (Fig. 10a and 10b) allow speculations about infilling mechanisms of giant landslide amphitheaters of volcanic origin. We find that high lava flow hazard areas are located mainly along the wall of the landslide scarp. Flows are then likely to follow pathways down the flanks, along the edges of the scarp. We would expect generally similar main lava flow hazard patterns for other ocean islands with infilling landslide amphitheaters and comparable topographic structure, such as Piton de la Fournaise (La Réunion, France) or Teide Volcano (Tenerife, Spain).

However, regarding the lava flow hazard estimation, we are left with uncertainties. The DOWNFLOW hazard map generation depends on the topography, $\Delta h$, the PDF of vent opening, and the lava flow length constraint. With our new post-eruptive DEM, we have a very reliable dataset for the DOWNFLOW simulation, at a resolution that meets, or even exceeds the requirements of the model. The parameter $\Delta h$ has a wide range of fit according to the calibration (Sect. 3.3.2, Fig. 4), meaning that the parameter $\Delta h$ is not among the main sources of error. As for the PDF, previous studies have shown that

even with a low number of vents, the resulting vent distribution is robust (Tarquini and Favalli, 2013). In contrast, the lava flow lengths constraint is known as a potential source for large errors (Tarquini and Favalli, 2013). In our Fogo case study the historic record is sparse, which causes the lava flow length constraint to be poorly defined. Overestimating the lava flow length would produce hazard maps with high hazard zones smeared, or extended downhill. Underestimating the lava flow length produces hazard maps with high hazard zones that shrunk uphill. To minimize the introduced error, we take into

account a rather large range of possible flow lengths.

Future studies are needed to address the question why a very similar pathway was reused by the magma during both, the 1995 and 2014-2015 eruptions. Therefore, we do not state that a future vent will open within the circle that is shown in Fig. 6 and Fig. 9b, at approximately the same location as during the two most recent eruptions. Rather we provide a hazard map for this scenario because at this point of time and knowledge, we cannot ignore the possibility that a future vent will be located

there. In the same way as the pre-eruptive hazard maps would have been useful to forecast the 2014-2015 lava flow paths, the post-eruptive hazard maps are valid for the next eruption of Fogo Volcano. Now that we know the exact location of the 2014-2015 vent, we can state that the pre-eruptive hazard map for the "Pico Pequeno scenario" (Fig. 9a) clearly predicts the lava flow path of this eruption, especially for the early phase of the eruption, with a very high level of confidence. This map can be treated as a forecast of the 2014-2015 lava flow path. However, due to limitations of the DOWNFLOW algorithm to

reconstruct lava flow emplacement of effusive pulses and lateral outbursts upon an initial update of the local topography by earlier lava flows, the general path agrees well, but the width of the lava flow is underestimated locally. Our results suggest that, if the next eruption occurs from Pico Pequeno, most likely, the initial lava flow path will be the one provided by the post-eruptive "Pico Pequeno" hazard map (Fig. 9b).

When comparing the pre- (Fig. 9a and 10a) and the post-eruptive hazard maps (Fig. 9b and 10b), the 2014-2015 eruption

changed the local lava flow hazard significantly at areas that are now covered by the 2014-2015 lava flow (general decrease), and along its edges (general increase). Even though the distribution of hazard has changed, the hazard at the village of Portela has not changed significantly in terms of the maximum probability of lava flow invasion. We observe a maximum



probability of lava flow invasion of 89 %, both, before and after the 2014-2015 eruption, but within the town the highest probability now exists for the parts of Portela that had low probabilities before the eruption and were also not covered by the lava flows (especially obvious in Fig. 10a and 10b). However, a new building was constructed on top of the 2014-2015 lava flow (indicated with the letter "H" in Fig. 9 and Fig. 10). The hypothesis that lavas are unlikely to inundate areas that were

previously overflown is a common assumption at Fogo (personal communication with residents). The new building "H" is located where the 2014-2015 lava flow is almost twice as thick as the average 2014-2015 flow thickness. Nevertheless, in this area the lava flow hazard remains high (Fig. 9 and 10, Sect. 4.4.2).

Our results (Fig. 9 and Fig. 10) also show that the village of Bangaeira is just as prone to be invaded by future lava flows as before the 2014-2015 eruption. Therefore, the assumption that previously overflown areas are now safe and won't be overrun

during the next eruption cannot be confirmed for this region either. The only chances for the villages to remain untroubled would be either to have an eruption from a vent outside the catchment (Fig. 11), or a vent inside the catchment but producing shorter lava flows as compared to the previously observed cases, or that early effusive pulses update the local topography in a way that our hazard maps are no longer valid.

Of the inhabited areas within the Chã, solely Ilhéu de Losna shows a significant (46 %) decrease in lava flow hazard. That

said, probability values out of range of the statistics are always possible, meaning that the next lava flow might also affect areas with very low probability.

Whenever lava flows threaten communities, debates arise regarding the control and diversion of lava flows in order to avoid economic loss (e.g. Barberi et al. 1993, Poland et al., 2016). Chirico et al. (2009) tested the effect of artificial barriers on probabilistic lava flow simulations. This way they found the most effective location, shape and orientation of a lava flow

barrier for the western part of the village of Goma, DRC, while they also concluded that the eastern part of the town could not be effectively protected. As shown by these and many other examples in the past, protection should be a matter of long-term investigations, rather than a rapid response once an eruption is already ongoing. Neither before, nor during the 2014-2015 lava flow crisis at Fogo Volcano, were artificial barriers built. As reconstruction of settlements upon destruction by basaltic lava flows is happening at Fogo Volcano and elsewhere in the world, the development of methods to better

understand the impact of man-made structures, including buildings and barriers, will continue to be a key task for future lava flow studies and simulations. Very high resolution topographic information, as available from TLS and photogrammetric surveys and modern, high resolution satellite systems, will facilitate such studies and help to significantly improve our understanding of lava flow emplacement processes and hazards.

## 6 Conclusion

The 2014-2015 Fogo eruption ended on 8 February 2015 after 85 days of activity, meanwhile three prominent villages were destroyed by lava flows. Satellite radar observations allowed time dependent mapping of the lava flow evolution during the 2014-2015 eruptive crisis of Fogo Volcano. Furthermore, we used combined TLS and photogrammetric data to quantify the thickness of the lava flows. We observed a maximum flow thickness of ~35 m at the vent; a remarkable thickness of up to ~25 m was observed approximately 4 km away from the vent in a ponding area west of the village of Portela. We estimate a

total erupted lava volume of $43.7 \times 10^6 \, \mathrm{m}^3$ +/- $5.2 \times 10^6 \, \mathrm{m}^3$ by comparing pre- and post-eruptive topographies. This eruption was therefore comparable in volume, covered area, and source location to the 1995 eruption.

Based on thousands of lava flow simulations we produced lava flow hazard maps for Fogo Volcano. These are provided to residents and decision makers for consideration when planning new infrastructure and the resettlement of the villages. The maps were produced using the DOWNFLOW stochastic model (Favalli et al., 2005) on the base of our high resolution

topographic map. Results of our lava flow hazard analysis imply that the two main villages within the Chã, Portela and Bangaeira, remain at high risk. Even the area west of Portela, where the topographic relief was partially infilled with a lava



flow of up to ~25 m thickness, is likely to be invaded again during a future eruption. These findings imply that an area once covered by a lava flow may again be overrun in the following eruption, at Fogo Volcano and elsewhere in the world. We also show which area of future vent opening will likely produce lava flows reaching and therefore affecting two villages (Fig. 11). The lava flow hazard at the location of Ilhéu de Losna, however, has significantly decreased. We conclude that the

5  next lava flow will very likely change the lava flow hazard within the Chã again. Our study therefore highlights the need of updating lava flow hazard maps shortly after eruptions in order to quantify the changes in lava flow lines. Updated topographic datasets are a key parameter for lava flow hazard assessments at volcanoes in general.



**Data availability**

We provide our hazard maps, the thickness map of the 2014-2015 lava flow, the catchment map and shaded reliefs of all used and generated DEMs in kml-format in the supplementary material of this publication. Also, all datasets are available online in geotiff-format (doi).

**Appendix A – Field observations**

During our field work between 11 January 2015 and 21 January 2015, Pico Pequeno displayed three different phases of activity: 1) Mild degassing from the vent, sometimes in combination with lava flows from the mount SW of Pico Pequeno (cf. Fig. C), accompanied by minor ash puffs without much explosive activity and no audible sounds. This activity was observed several early mornings. 2) Distinct ash puffs every 0.5 to 5 minutes of up to 700 m height. These explosions emit

bombs which mostly fall back into the vent. A sloshing sound can be heard from the lava within the conduit and explosions are sometimes so loud, that an echo from the Bordeira can be heard. This activity was mostly observed in the afternoon. 3) Intense explosive activity, almost continuous high lava fountains (up to 300 m) well above the crater rim. Explosions eject bombs up to 700 m, impacting half way up Pico and 2/3 down Pico Pequeno. Ash explosions rose up to 1,000 m, were then drifting above Pico and raining out mostly on its flanks. Intense, almost constant noise, with many echoes from the Bordeira

was audible. This activity was observed on four different days around sunset (18:30 pm to 19:00 pm LT (UTC-01:00)).

**Appendix B – Masks**

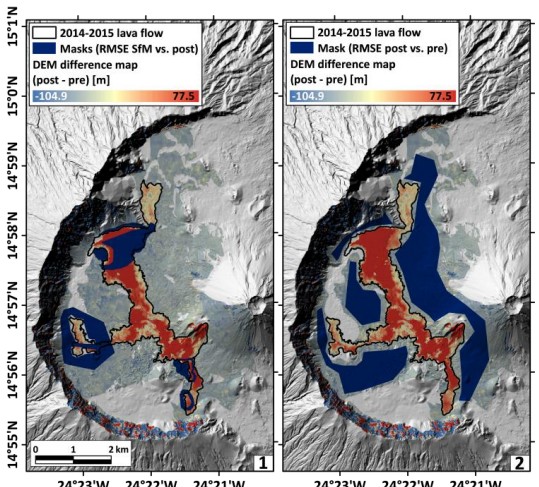

**Figure B: Minuit RMSEs (1) in the blue regions (SFM DEMs), from top to bottom: 0.75 m, 1.45 m, 1.79 m, 0.69 m and (2) in a "buffer" area outside the lava flow (post- vs. pre-eruptive topography): 0.81 m.**

**Appendix C – Multitemporal TLS analysis**

A small lava flow was still active during our field work on 12 January 2015. In order to map possible changes in the flow field, including active lava flows, the set-up on Monte Saia was kept the same over the duration of our field campaign, i.e. 6 reflectors and the scanner tripod were permanently installed (see Figure C). From this position (see scanner position in Figure C and SAIA1 in Table C) we acquired three multi-temporal, very high-resolution 360° scans (on 12 January 2015, 17

January 2015 and 21 January 2015, respectively). Details on the tie point registration and Minuit minimization results of these three scans are listed in Table C.




Only the first of these scans was used in combination with the Beco and Amarelo scans to produce the updated DEM (due to slightly worse weather conditions on the other days). Therefore, the active lava flow is not completely included in our post-eruptive DEM, which is only updated until 16 January 2015 (the acquisition date of scans MBC 4 and MBC 5, cf. Table 1). Even though the TLS points acquired on 17 January 2015 and 21 January 2015 were too sparse to create a DEM, they were

5    dense enough for estimating the flows' volumes to roughly amount to $0.15 \times 10^6 \, \text{m}^3$ (from the vertical difference between SAIA3 and SAIA1, cf. Fig. C1) and $0.05 \times 10^6 \, \text{m}^3$ (from the vertical difference between SAIA3 and SAIA2, cf. Fig. C2).

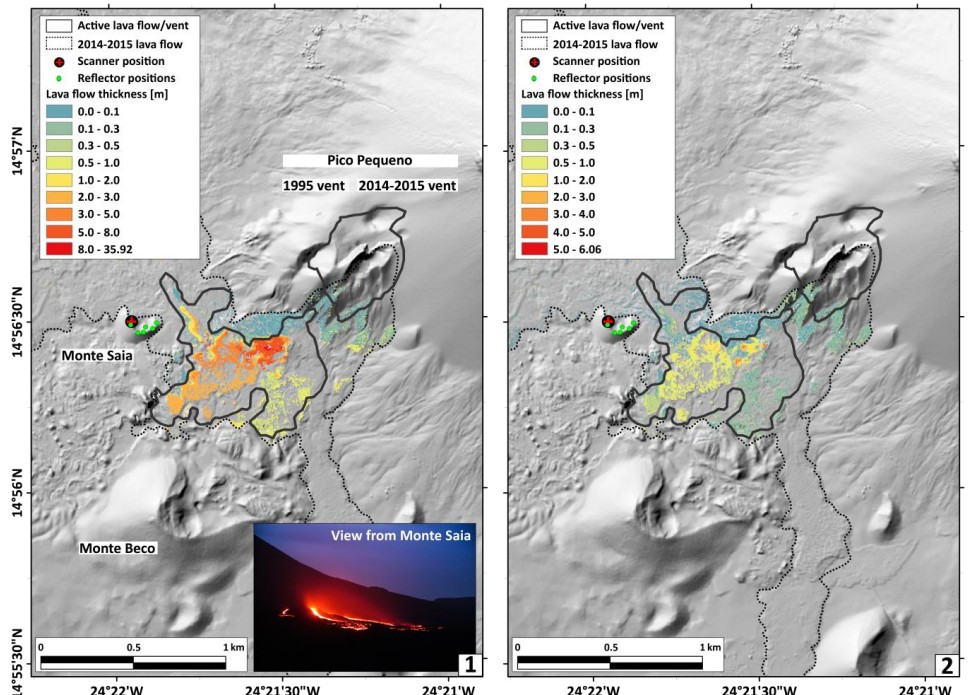

**Figure C: Thickness maps of the active lava flow, calculated from the vertical difference between the TLS data acquired on 21 January 2015 minus (1) the TLS data acquired on 12 January 2015 and (2) the TLS data acquired on 17 January 2015 (all in grid**
10  **format). Outlines of the lava flows are taken from TerraSAR-X coherence (Sect. 3.1, 4.1, 5.1 and Fig. 7). Inset shows a photograph taken in the early morning of 12 January 2015 from a position on Monte Saia, looking southeast.**

**Table C: GPS coordinates of scanner position and point cloud accuracies of the multi-temporal TLS data acquired from Monte Saia.**

| Scan Name | Scanner Location (GPS) | | | | | | Date | Local Time | #Points | #Points (filtered, cleaned) | Tie Point Registration | | Minuit | |
|---|---|---|---|---|---|---|---|---|---|---|---|---|---|---|
| | Lat | Lat Error | Long | Long Error | H | H Error | | | | | #Tie Points | St. Dev. [m] | #Points | RMSE [m] |
| SAIA1 | N14°56'29.86298'' | 0.003 | W24°21'56.92498'' | 0.003 | 1856.169 | 0.008 | 12.01.2015 | 09:45 | 98068086 | 20027376 | 6 | 0.0064 | 6921595 | 1.320337 |
| SAIA2 | | | | | | | 17.01.2015 | 20:30 | 55767098 | 13325875 | 6 | 0.0033 | 4712745 | 1.251083 |
| SAIA3 | | | | | | | 21.01.2015 | 11:05 | 54885049 | 11611177 | 6 | 0.0048 | 3943593 | 1.266489 |

**Author contribution**

15   NR, EdZ, and JL carried out the field work. SV provided assistence in the field and contributed photographs. NR processed the TerraSAR-X data. MF developed the DOWNFLOW model code. MF and NR performed the TLS and SfM data processing and model simulations. RF processed the GPS data. TRW and AF helped with the photogrammetric data processing. NP provided the pre-eruptive DEM. NR prepared the manuscript with contributions from MF. TRW, AF, EdZ, RF, and SV helped improving the manuscript.



**Acknowledgements**

This work was supported by the Helmholtz Alliance "Remote Sensing and Earth System Dynamics" (HGF EDA). Field work was funded by the GFZ HART program. Access to the Riegl instrument was kindly provided by Niels Hovius, Head of Sect. 5.1 of the GFZ. Operating instructions were given by Kristen Cook. The C4G monitoring campaign was made possible

by an emergency financial support provided by Fundação para a Ciência e Tecnologia, Portugal. We thank Eleonora Rivalta, Jacqueline Salzer and Adam Mehlhorn for valuable suggestions which helped improving the manuscript. We also thank Paulo Fernandes Teixeira and Lourenco Francisco Fernandes for their invaluable assistance during fieldwork on Fogo Island.

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





**Tables**

**Table 1. GPS coordinates of scanner positions and accuracies of point cloud combinations.**

| Scan Name | Scanner Location (GPS) | | | | | | Date | Local Time | #Points | #Points (filtered, cleaned) | Tie Point Registration | | Minuit | |
|---|---|---|---|---|---|---|---|---|---|---|---|---|---|---|
| | Lat | Lat Error | Long | Long Error | H | H Error | | | | | #Tie Points | St. Dev. [m] | #Points (used) | RMSE [m] |
| SAIA1 | N14°56'29.86298'' | 0.003 | W24°21'56.92498'' | 0.003 | 1856.169 | 0.008 | 12.01.2015 | 09:45 | 98068086 | 20027376 | | | 6921595 | 1.320337 |
| MBC 1 | N14°55'57.45007'' | 0.001 | W24°21'52.20829'' | 0.002 | 1961.930 | 0.004 | 13.01.2015 | 17:05 | 115648260 | 27390544 | 3 | 0.009 | 14617678 | 1.396584 |
| MBC 2 | N14°55'56.67938'' | 0.002 | W24°21'52.45296'' | 0.002 | 1966.088 | 0.005 | 13.01.2015 | 20:00 | 128026401 | 24656260 | 3 | 0.0025 | 11896245 | 1.419237 |
| MBC 3 | | | | | | | 14.01.2015 | 11:30 | 68285868 | 14739264 | 4 | 0.0045 | 7188509 | 1.034169 |
| MBC 4 | N14°55'56.67938'' | 0.014 | W24°21'52.74932'' | 0.014 | 1968.339 | 0.046 | 16.01.2015 | 13:10 | 147229401 | 18912861 | 3 | 0.0086 | 8897666 | 1.397200 |
| MBC 5 | N14°55'55.75526'' | 0.007 | W24°21'52.55238'' | 0.008 | 1968.706 | 0.025 | 16.01.2015 | 17:05 | 128026401 | 20515900 | 4 | 0.0062 | 10803615 | 1.068116 |
| MBC ALL | | | | | | | | | | 106214829 | | | 53402388 | 1.299546 |
| PORT1 | | | | | | | 17.01.2015 | 15:00 | 123906195 | 19058183 | 4 | 0.0058 | | |
| PORT2 | | | | | | | 17.01.2015 | 16:35 | 140828401 | 19496082 | 4 | 0.0058 | | |
| PORT ALL | | | | | | | | | | 38554265 | | | 23913754 | 1.355356 |
| FINAL CLOUD | | | | | | | | | | 164761510 | | | 80609498 | 1.077789 |

**Table 2. Acquisition and processing of SFM data.**

| SfM Patches | Date | Camera | # Images | # Points | # Points used (mask) | Minuit RMSE |
|---|---|---|---|---|---|---|
| NW lobe | 18.01.2015 | Canon EOS REBEL T3i | 12 | 1134322 | 834649 | 0.750539 |
| S lobe (north) | 19.01.2015 | Canon EOS REBEL T1i | 48 | 431340 | 29219 | 1.791653 |
| S lobe (south) | 19.01.2015 | Canon EOS REBEL T1i | 48 | 431340 | 27252 | 0.694206 |
| W lobe | 18.01.2015 | Canon EOS REBEL T1i | 17 | 1003348 | 251517 | 1.452028 |





**Figures**

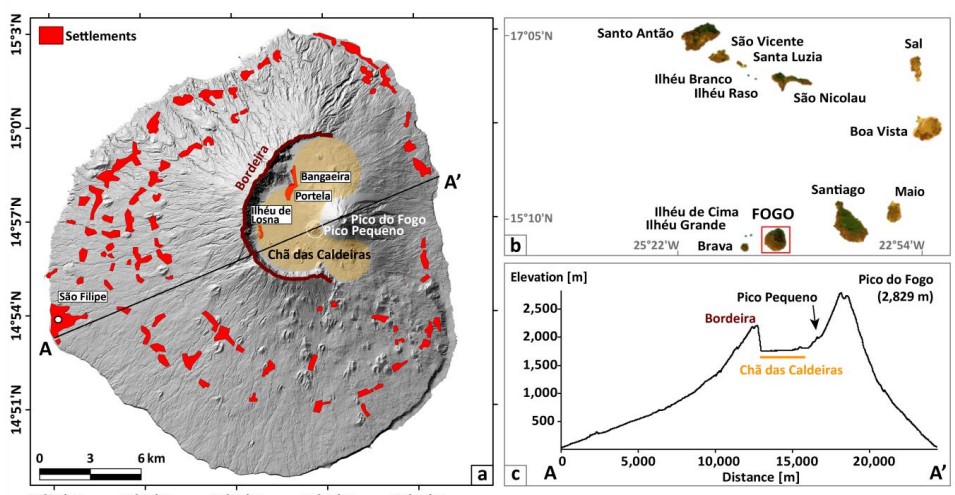

**Figure 1: Map of Fogo Island (a), as one of the 14 volcanic islands of the Cape Verdean Archipelago (b), and profile A – A' cutting through the lowest point of the Bordeira wall (c). The maximum elevation of a point along the Bordeira wall is 2692 m. The layer**
5 **of settlements (red areas) was downloaded from the Copernicus Emergency Management Service (2014), the small settlement of Ilhéu de Losna was added manually to the map.**

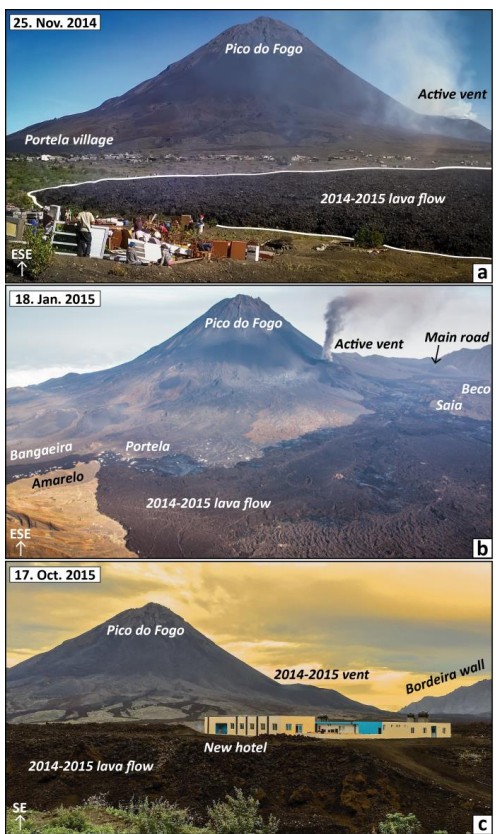

**Figure 2: Photographs taken during (a and b) and after (c) the 2014-2015 Fogo eruption. On 25 November 2014, the village of Portela was not yet engulfed by lava flows (a), while on 18. January 2015 lava flows were covering the villages of Portela and**
10 **Bangaeira (b). In early October 2015, a new hotel opened which is built on the 2014-2015 lava flow (© Mustafa Kerim Eren) (c).**




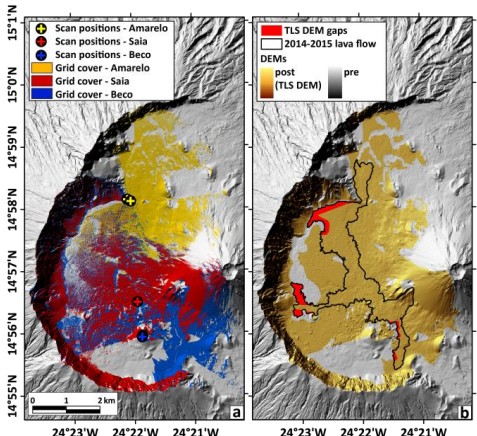

**Figure 3: Coverage of the TLS datasets (in grid format) and corresponding scanner locations on Monte Saia, Monte Beco, and Monte Amarelo (a) and coverage of the final TLS DEM (b), where red shapes mark areas of the 2014-2015 lava flow (black polygon), that are not covered by the final TLS DEM. These gaps were filled using camera data analysed with the SfM method.**

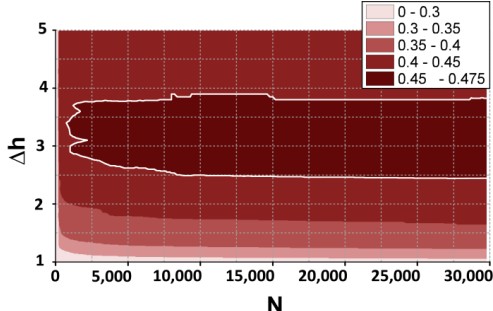

**Figure 4: DOWNFLOW calibration. The maximum fitness of $\mu = 0.47$ is achieved using a parameter $\Delta h = 3$ m. We chose $N = 10,000$ as this guarantees short computing times while maintaining statistical robustness.**

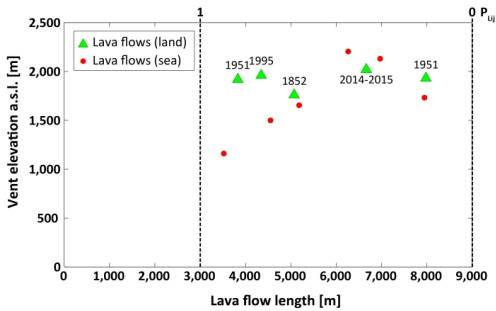

**Figure 5: Lava flow lengths constraint. Green triangles show the lava flows that stopped on land, while red points indicate lava**
10 **flows that reached the ocean.**

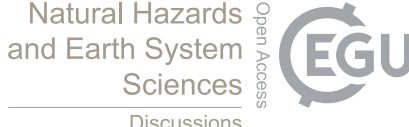



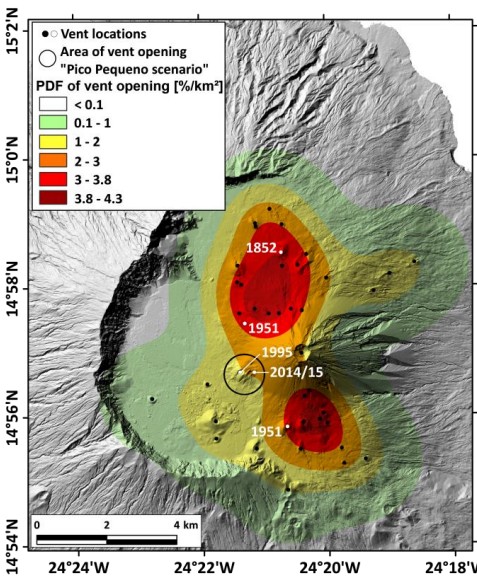

**Figure 6: Probability distribution of future vent opening for the "Fogo scenario". White points mark the locations (and eruption years) of historic vents producing lava flows that stopped on land. Black dots mark locations of historic vents producing lava flows that reached the ocean. The black circle around Pico Pequeno indicates the PDF domain for the "Pico Pequeno scenario".**









Figure 7: TerraSAR-X coherence maps (georeferenced, i.e. north is up). Panels a, c, e, and g are scenes acquired along orbital path 57 (ascending), panels b, d, f are ascending acquisitions (track 148), and panel h is a coherence map between two SAR images acquired along the descending track 155.





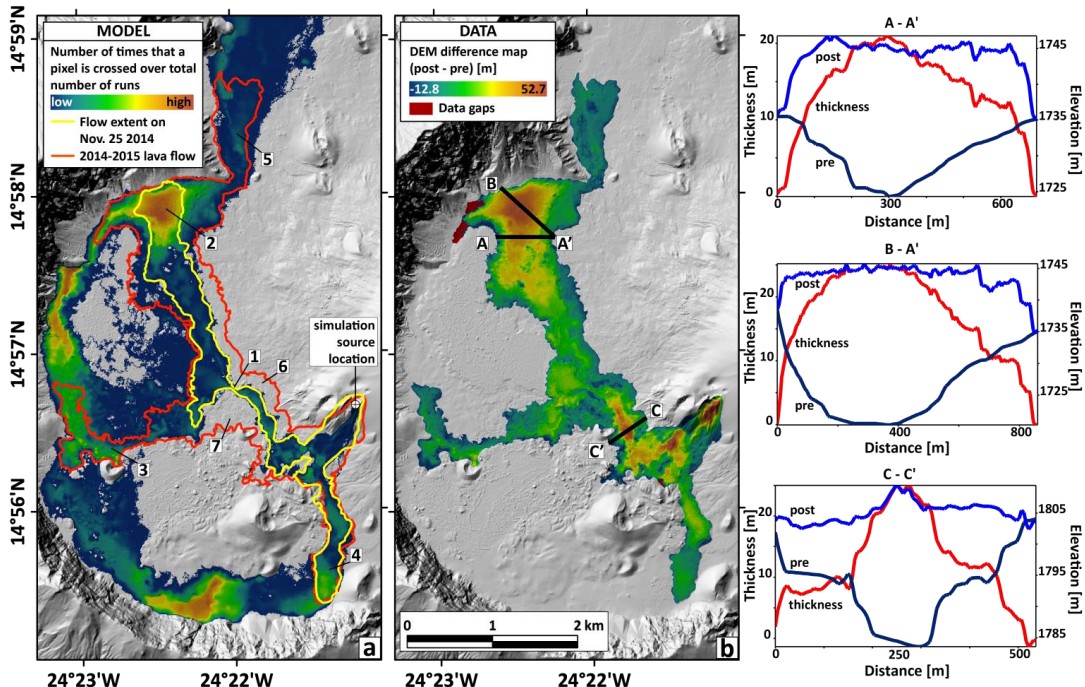

**Figure 8: Comparison of the DOWNFLOW reconstruction of the 2014-2015 lava flow (a), and the lava flow thickness (DEM difference map) (b). Profiles compare pre- and post-eruptive topographies and the lava flow thickness.**

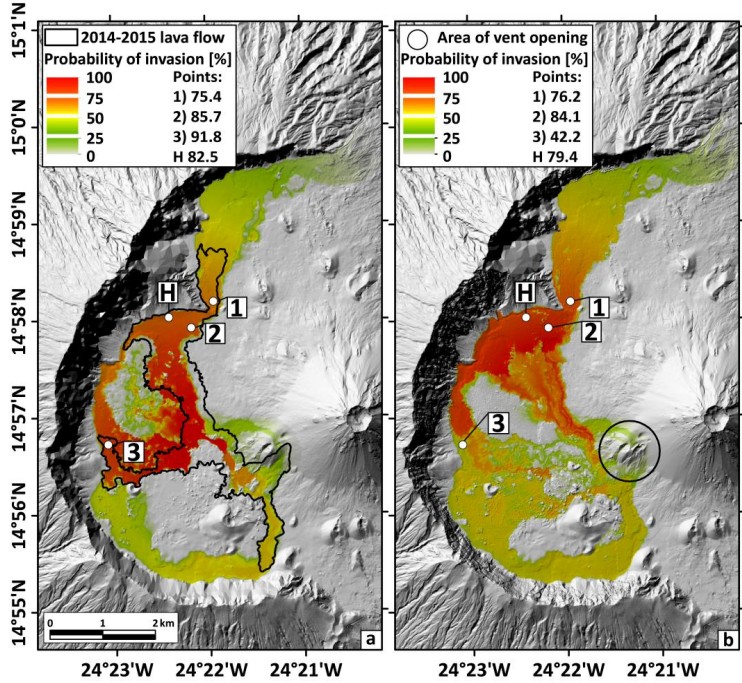

5   **Figure 9: Pre- (a) and post-eruptive (b) hazard maps for the "Pico Pequeno scenario". Numbers of points correspond to the locations of the villages: 1) Bangaeira, 2) Portela, and 3) the settlement of Ilhéu de Losna. The letter "H" marks the location of the new hotel which was built on the 2014-2015 lava flow.**





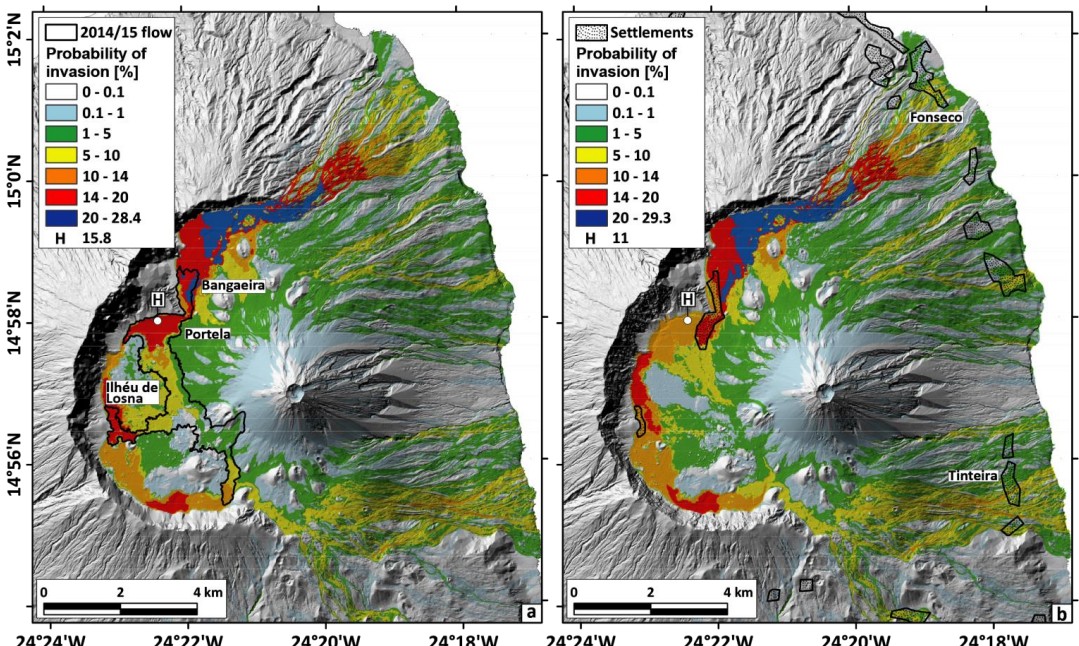

**Figure 10: Pre- (a) and post-eruptive (b) hazard maps for the "Fogo scenario".**

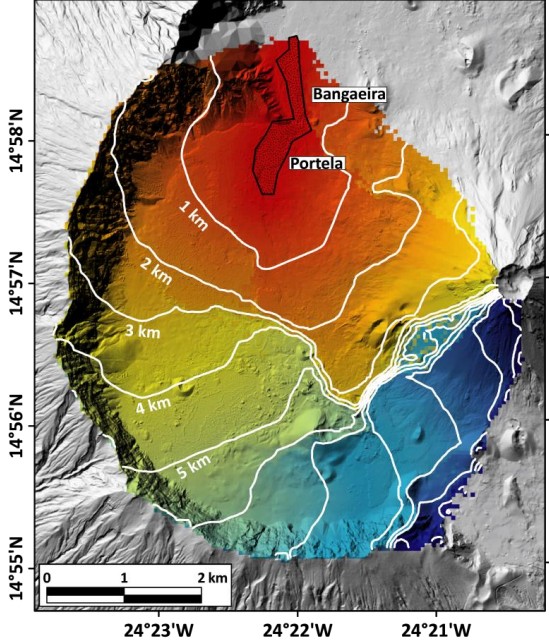

**Figure 11: The coloured area shows all locations of future vents that will likely produce lava flows reaching and therefore affecting the villages of Portela and Bangaeira. Vents are herein ranked according to the minimum length that lava flows need to travel to reach the villages.**