# Peer review of "Lava flow hazard at Fogo Volcano, Cape Verde, before and after the 2014-2015 eruption"

_Natural Hazards and Earth System Sciences, 2016_

## Referee Comment (RC1) · M. P. Poland (Referee) · 5 Apr 2016

The manuscript "Lava flow hazard at Fogo Volcano, Cape Verde, before and after the 2014-2015 eruption," by Richter and others, makes use of detailed topographic and surface area data to map lava flow inundation and assess hazards using the stochastic DOWNFLOW model. An exceptional amount of data are used in this analysis—the post-eruptive DEM required intensive field work with a TLS instrument in addition to the collection of a photogrammetric dataset. Despite the complicated nature of the topographic data, which span several orders of magnitude in terms of resolution, the analyses are well executed and demonstrate the synergy between rapid collection of topography and currently available lava flow pathway models. The DOWNFLOW simulations for before and after the 2014–2015 eruption yield important insights into hazards within the Cha das Caldeiras and have great relevance today, with the potential

for application to land-use planning to minimize the risk from future eruptions. There are general lessons that can be applied to other basaltic volcanoes as well.

The writing is excellent, the figures are of high quality, and the paper is well referenced. The details and conclusions not only offer a great description of the 2014-2015 Fogo eruption, but also the complexities of lava flow hazards and human responses. I have only a few thoughts of consequence that I'd like to share with the authors. I also have a number of minor comments that I list at the end of this review and that mostly point out places where the text could be reworded for clarity.

I hope that the authors find this review helpful.

Best wishes, Mike Poland

Specific comments:

- I would have liked to see a little more background about Fogo's eruptive history, or perhaps a reorganization of the material that is presented. For example, in the introduction, the 1951, 1995, and 2014-2015 eruptions are mentioned first, and then the "major 1680 eruption" is brought in somewhat casually. This sounds like a key event in the volcano's history, yet its importance is hard to appreciate in the current context. It's not even mentioned in the "Geologic setting and eruptive history" section. I recommend that the entire second paragraph if the introduction (which starts with "Fogo Island features a prominent giant landslide") be integrated with the "Geologic setting and eruptive history" section. That way, the introduction covers only that material which is sufficient to understand the purpose and importance of the paper, and all of the information on past eruptions and geologic setting are contained within a specific section.

- In the introduction, I found it a little odd to see that "DOWNFLOW is known to work well at steep terrain" and then in the subsequent sentence "Here we apply the model to a rather flat area." This seems contradictory. Is this one of the first studies to apply DOWNFLOW to a low-relief area? If so, is that an outcome of the paper—that the

model is also appropriately used where topography is subdued (assuming the model parameters can be adequately explored and constrained)?

- There are a number of very short sections throughout the manuscript, and it might be better to eliminate or merge some of these just to keep from breaking the flow of the paper. For example, it seems to me that section 3.3.1 could be merged with section 3.3.2. It keeps the description of the DOWNFLOW simulation with no length constraint contained within a single section, thereby lessening the chance for later confusion (in my opinion). Also, section 4.4 (the single sentence before section 4.4.1) is not really necessary and could be deleted.

- I enjoyed the discussion about the challenges of placing length constraints on the lava flow simulation. That said, I was a little surprised there was no mention of the fairly well-known work of Walker (1973) and subsequent authors, who noted the correlation between effusion rate and flow length. This could be especially important given the restricted vertical spread of historic eruptions at Fogo (making it impossible to infer a correlation between flow length and vent elevation). Given that one of the really noteworthy results of this work is a volume for the 2014-2015 lava flow, the authors could calculate discharge rates. Overall, the eruption rate (43.7 MCM over 85 days) was 6 m3/s, but it was certainly higher during the first few days/weeks of the eruption, when the flow traveled farthest. Can the authors place some constraint on the maximum effusion/discharge rate for the early part of the eruption? This might allow them to use Walker's (1973) relation between effusion rate and flow length, and could also facilitate a more thorough comparison with the 1995 eruption.

- What is the impact of varying delta-h on the simulations? That parameter seems relatively well constrained, but a brief mention of how important that value is to the results (what happens if the value is changed to 3.5 m? Or 2 m?) would help reinforce the simulation results.

- I was struck by the fact that Pico Pequeno, where the last two eruptions have taken

place, is not the densest concentration of vents—not by a long shot! This is remi-
niscent to me of the situation at Campi Flegrei, where Bevilacqua et al. (2015) found
that the most likely location of a new vent in that caldera was not where the most re-
cent eruption occurred. This has little direct relevance, except with underscoring the
importance and challenge of knowing something about vent location when estimating
hazard.

- I really like the coherence maps showing lava flow area, but wonder if those might
be summarized in a single figure? The maps dominate the figures (in terms of their
size and the space they take up), even though they are not necessarily the most critical
figures in the paper. One option might be to move the coherence images to the ap-
pendix, and replace the separate panels of Figure 7 with a single map that shows the
flow outline at different times (because there is no look angle artifact with coherence,
different dates could be combined on a single map). This would make it easier to com-
pare what are now separate panels and show the spatio-temporal growth of the flow
at a glance. A disadvantage is that one would not necessarily "see" the small flow that
was active in mid-January, since it is mostly contained within the existing flow. That's
why the coherence images could remain in the appendix, but the areal growth of the
flow could be represented in a single manuscript figure that would communicate that
information more succinctly. Also, why do the coherence images not go through the
end of the eruption, in early February? If they show no change in the flow (aside from
incoherence due to cooling and rapid subsidence), that would be helpful in constraining
the effusion rate (i.e., that there was little volume erupted during late January – early
February). This is already hinted at in section 4.2, but there is little evidence given to
back up claim of a post-January 16 erupted volume of 0.05 MCM.

- The manuscript does a good job of pointing out that topography must be updated
in order for subsequent DOWNFLOW runs to be relevant, and indeed, this is borne
out by the results—some areas had a 0% chance of inundation based on the pre-
eruption DEM, but as the 2014-2015 eruption progressed, the modified topography

caused these areas to become inundated. I think this could be stated more explicitly, however. Lava flows make their own topography, so the failure of the simulation to exactly predict the flow coverage (the "0%" probability areas) is really more a reflection of the changing topographic conditions than of the capability of the simulation, right? This is a major conclusion of the paper, but it is somewhat spread out between sections 4 and 5. The authors might want to make it more explicit. Perhaps even address the question about iteratively updating topography (with TLS?) during a crisis so that successive DOWNFLOW runs will be more representative of changing conditions. That seems like an unstated conclusion, given the datasets the authors used and the simulations they ran.

- Was any adjustment applied to the volume calculation to account for vesicularity? For aa flows, ∼25% is often assumed. If no adjustment is made, the authors should be explicit that the volume is a bulk volume, and that vesicularity is not accounted for. This might explain a small percentage of the difference with the Ferrucci volume...

Technical recommendations:

- I suggest deleting from the abstract the sentence "Based on this, we discuss how our study can help improving the general understanding of basaltic lava flow behavior." Such a discussion really doesn't exist in the manuscript, and I don't think anything is lost from the abstract if this sentence is removed (in fact, the abstract becomes tighter).

- The authors reference "Harris and Rowland, 2001" in the introduction section to highlight their FLOWGO code, but maybe referencing their 2015 update to that code would be better? That paper is included in the AGU monograph on Hawaiian volcanism.

- Toward the end if the introduction, it is unclear whether "featuring a 5 m spatial resolution" is referring to the pre-eruption DEM, the post-eruption DEM, or both.

- Is the last paragraph of the introduction ("In the first section of this paper...") necessary? I think the introduction might be more powerful if it were to end with mention that

this work is the first to use TLS data as a base to develop probabilistic hazard maps.

- In the first paragraph of section 2, note that Bordiera reaches a height (not an elevation) of 1,000 m above the Cha.

- The first sentence of the second paragraph of section 2 is awkward, and should be rephrased. It implies that reports of eruptive activity exist for the period ~1500-1660, but afterwards there was less information? This lesser-known period would include the time period of the "major 1680 eruption."

- At the end of section 2, I suggest replacing the word "reenacted" with "renewed."

- In section 3.1, the phrase "ranges between ~12 h (for the ascending and descending pairs 57/64 and 148/155) to ~6 days" is awkward and should be reworded. I think the authors mean that consecutive ascending and descending data are offset by 12 hours, but the two sets of A/D pairs are offset by about 6 days. In any case, this might be confusing to readers who don't regularly work with TSX data. Also in this section, it might be useful to mention that vegetation is not an issue in the Cha, so coherence really just reflects steep slopes and surface change. This is stated much later in the manuscript, but should probably also be explained here.

- In section 3.2.2, I was a little confused by the scanner locations and positions. It appears that there were three major locations from which scans were collected—Beco, Saia, and Amarelo—and at two of these locations, multiple positions were occupied (presumably with different fields of view). Is that right? After the three locations are mentioned, it is stated that "At five of the scanner positions" GPS data were acquired. It is not clear if these 5 positions are distributed between the Beco and Amarelo sites, or represent all of the positions at these sites, etc. I recommend that a little rewording be done here to make the procedure easier to understand.

- Toward the end of section 3.2.2, the phrase "here used methodology" is awkward and should be reworded ("methodology used here" would be fine).
- In section 3.4, I would delete mention of the "~82,000 vents," since it only raises the question of how that number was determined. Since this is explained in section 3.4.3 in greater detail, the earlier mention can be removed.

- In section 3.4.2, the phrase "which integrates up to be 1" should probably be reworded to something like "which sums to 1."

- In section 3.4.3, right after equation 2, "extents" should be changed to "extends."

- In section 4.1, did the flow thicken after it stopped advancing (after December 23, 2014)? From figure C, it looks like it did (at least, the active lobe in January did), and that would probably be worth stating directly.

- In the first paragraph of section 4.2, I didn't understand what was meant by the error being smaller "when comparing post-eruptive and pre-eruptive grids".

- In the second paragraph of section 4.2, isn't the area calculated from the coherence maps, and not the topographic difference? Also, note that the area given here (4.84 km2) differs from the area given at the end of section 4.1 (4.85 km2). Finally, perhaps the maximum thickness of the 2014-2015 flow could be given along with the average thickness, instead of at the end of the section?

- In section 4.3, I thought it was a little awkward to bring up the apparent correlation between the simulation and the thickness, since it is not raised again until well into the discussion. Maybe wait until the discussion to note this similarity? That way, it doesn't get in the way of the description of the simulation results.

- The first time a percentage is given in terms of a DOWNFLOW result (in section 4.4.1), it might be useful for the authors to offer a brief explanation of what that percentage is referring to—for example, is it the likelihood that a future eruption will inundate a specific pixel? Just so that the reader is clear on the meaning.

- Toward the end of section 4.4.2, "remarkable hazard" is an awkward phrasing that should be reworded. It's unclear if 10% is remarkable because it is so low or so high.

- The idea of "catchment" maps is a good one. The authors may wish to reference some work along the same lines at HVO by Frank Trusdell and Jim Kauahikaua, who use that technique for hazard assessment on the Island of Hawaii.

- At the end of section 5.2, is there any indication why the volume derived from the topographic difference is so much greater than that of Ferrucci et al., 2015? The idea that volumes inferred from thermal data might be so much different from those determined by topographic differences is a little unnerving.

- In the third paragraph of section 5.4, the authors raise the question of why the 1995 and 2014-2015 flows followed such similar paths. But isn't the answer "topography"? Can it be anything else? It's unclear to me what type of "future studies" might actually address this question.

- In the conclusions section, I would recommend deleting the second-to-last sentence. "We conclude that the next lava flow will very likely change the lava flow hazard within the Cha again." This sounds rather grand, but is also pretty plain for all to see, and the point is made more effectively earlier.

---

## Referee Comment (RC2) · S. Calvari (Referee) · 7 Apr 2016

The paper is very nicely written, clear, novel and informative. It is extremely useful for the present evaluation of volcanic hazard from lava flow invasion within the Cha Caldera on the summit of Fogo. The authors presented a clear and useful analysis of their DEM data. I have only minor edits corrections, but look forward to see the paper published in its final form.

Please also note the supplement to this comment:
http://www.nat-hazards-earth-syst-sci-discuss.net/nhess-2016-81/nhess-2016-81-RC2-supplement.pdf

2016.

**Supplement:**

[revised manuscript text omitted]

---

## Referee Comment (RC3) · M. Kervyn (Referee) · 9 May 2016

Review of **Lava flow hazard at Fogo Volcano, Cape Verde, before and after the 2014-15 eruption** by Richter et al.

The study of Richter et al. presents an interesting analysis of the Fogo 2014-15 lava flow eruption. The two main point of the papers are 1. The presentation of the methodology to update the topography of the DEM using ground-based laser and photogrammetry, leading to the derivation of the lava flow volume, and 2. The impact of the new topography on the spatial distribution of lava flow modeled using a probabilistic approach.

The manuscript is of overall very good quality. It is well written and supported by informative figures and tables. The text is however quite long and could be shortened by 15-20% without loss of content, avoiding unnecessary broad information in the introduction, moving some technical details in appendix and avoiding the many repetitions between the observations and the discussion.

I recommend that this manuscript be accepted for publication in NHESS after minor revision of the text in order to address the comments listed hereafter.

**Major comments**:

1. My main concern is that the authors stress throughout the manuscript the relevance of their study for risk management, the rapid acquisition strategy of their method and the role of the HART of GFZ in assisting local decision makers during the crisis management and in training local residents. Although I do not doubt that the high quality products presented might be relevant for crisis and long term risk management, it is unclear in the manuscript to which extent the local actors were indeed informed about these results and how much these maps were used to inform the local population. Either the authors have taken actions to ensure that the scientific results have a direct impact on risk management and information to local population, and they should describe it, or they have not (yet) done so and assume that a scientific publication is sufficient to have an impact. In the later case, the argument of the impact of the research for risk management should be downscaled in the paper (eg. page 2, lines 1-6, page 12 lines 5-10, page 15 lines 37-38). As pointed out in the introduction, residents tend to re-occupy zones invaded by lava flows but from experience I don't believe accurate lava flow hazard maps can make a difference without major investment in education and communication actions. This is mentioned by authors (page 2, line 6) but this should be clear in the discussion and conclusion.

2. DOWNFLOW: I am quite familiar with the approach and capabilities of the DOWNFLOW code. In some places the authors should be more careful in the description of the DOWNFLOW results and be critical. On p 10 (lines 4-10), authors highlight the good match between the DOWNFLOW simulation and the outline of the actual flow. Where this is true for some areas (points 1,2, 3, 4 on Fig. 8), this is not so true for zone 5 where the probabilities of DOWNFLOW are much lower than other zones located at shorter distance (Western and Southern border of the calder). Zone of overestimation of the DOWNFLOW simulations

should also be described in this part of the results. Also in the discussion (section 5.3), authors should not only highlight the capabilities but also the limitations: between the Northern and NW branches, a lot of pixels have a low probability of lava flow invasion but were not invaded, whereas the opposite is true for Zone 5. These uncertainities, and their cause, should be highlighted. When discussing accuracy (page 13, line 31), quantitative values should be provided: the reader should be informed that 'very good' simulation have accuracy parameter of ~0.5 even without considering the issue of length.

**Minor comments:**

- Page 3 , line 7: spell out TLS when first used in main text
- A recently published paper by Cappello et al. (2016, JGR, DOI: 10.1002/2015JB012666) also discuss lava flow modelling for the Fogo 2014-15 eruption. As that publication use a physically-based model, a comparison of the advantage and limitation of the two approaches in the discussion would be useful.
- Paper by Albino et al. (JGR, 2015) presents a volume estimate for the Nyamulagira 2011-12 lava flow eruption using a TanDEM-X DEM. The author could compare the accuracy of their DEM comparison and volume estimate with the one presented by these other authors. Why was the TanDEM-X technology not applied in the case of the Fogo eruption to derive the post-eruption DEM?
- The authors argue that ground-based technology are 'more flexible' (page 3, line 5): I find this argument a bit weak, as ground-based technology require to access inhospitable volcanic area during or directly after an eruption. This is also contradicted by the discussion where the TLS approach is described as 'time consuming and challenging' (page 13, line 10).
- HART: authors repeatedly highlight the action of GFZ-funded HART initiative (page 4, line 15-18; page 12 line 3-10) and the rapid-response character of the action (page 4, line 16). Although HART is for sure a nice initiative I don't think this paper should aim at giving so much publicity for it. The author should also justify why they consider that the topographic survey is part of a 'rapid response': as eruptions are often separate by several years, this survey could be done once it is clear that the eruption is finished and access to the site is secured.
- Section 3.2.4: mention already here the spatial resolution of the post-eruption DEM produced
- 'filling algorithm' : in the methodology (page 6, line 33; page 10, line 10), as well as the discussion section (page 13, line 33), the authors refer to a 'filling algorithm' integrated in DOWNFLOW. More information should be given about this aspect. DOWNFLOW being a probabilistic model with no explicit lava flow thickness and topography adaptation, I don't really understand what is meant by 'filling algorithm' except for the possibility of the simulation to continue beyond actual pits in the DEM. How this is implemented in the algorithm should be explained in details. The observation that higher probabilities are found in pits, corresponding to thicker flow accumulation, although interesting, is not a surprise as it is a simple results of the topography-control on the lava flow paths. I would not say that these similarities are 'intriguing' (page 10, line 5), they are rather expected and logical based on the modelling approach.

- Section 3.3.2: the optimal Δh value is defined based on the maximization of the best fit parameters, using the actual lava flow as reference point. In the discussion, authors argue that this parameter has a wide range of fit (page 14, line 19), although figure 4 actually suggests that for Δh <2.5 m and >4 m, the fit significantly reduces. How confident are the authors that this Δh value will also be optimal for the future eruption?
- Section 3.4.1. : Authors should clarify that they assume a linear decline in probability from the minimum to the maximum length, similarly to previous application of DOWNFLOW. Bonne et al. (Int. J. Remote Sensing) demonstrated that for Mt Cameroon, a Gaussian probability decrease better fitted observed lava flows' lengths.
- Page 8, line 2: justify the bandwidth of the Gaussian kernel. This bandwidth can have a major impact on the resulting PDF map.
- Equation 2: explain why the resolution of the DEM Δx and Δy need to be taken into account in this equation.
- Fig. 8a: why is the color bar of this figure not presented in a quantitative way (with percentage) similarly to Fig. 9 and 10. This would be much more interesting. Actually the color scale of Fig. 10 is the most interesting one, as it also enable to know what is used as lower threshold (pixel with no color).
- Fig.8b: this is a key results of the study which could be better valorized. An histogram of the thickness distribution should be provided. The color bars suggest values from -12.8 to +52.7 m: what proportion of the thickness are negative, how could this be explained and how does this impact the total volume? Author mention a maximum thickness value of 35m: why does the color bar goes to 52 m then?
- Section 4.2: the aeral coverage of the lava flow on lines 18 and 28 (page 9) are not matching. Please provide an histogram of the thickness values (at the moment you give the mean and some max value, but you don't provide the distribution nor the minimum values that are probably below zero: Fig. 8b).
- Page 11: the pre-post hazard map comparison is interesting at the caldera scale for the 'Pico Pequeno' scenario. It is less relevant for the 'Fogo scenario' as the changes are minor and similar to the ones observed in previous scenario. I advise to shorten or cut lines 11-18 (page 11).
- Page 12, section 5.1: this section could be largely reduced as it brings little new information. The technique of coherence loss to map new volcanic products is indeed quite standard and does not disserve a long discussion. I disagree with the sentence (line 26-): *as the resulting extent of the mapped and the simulated 2014-15 lava flows match almost perfectly".* Looking at fig. 8a it is obvious that this is correct only for specific zones andonly for the early emplacement stage of the lava flow.
- Page 15, lines 17-28: this paragraph does not relate at all to the presented results. Although I know that the modelling approach might enable to simulate the influence and optimal location of a barrier, this is not done in this case, and I doubt this would be practical solution for the Cha caldera, since the eruption probability and the settlements are dispersed. The example of Chirico et al. (2009) mentioned is a good example of a purely theoretical modelling exercise with no applicability on the ground. I would advice to cut this paragraph.

---

## Author Comment (AC1) · 19 May 2016

We very much appreciate the detailed and constructive comments provided. We incorporated all suggestions into the manuscript; however, some remain to be fully included in the final version of the manuscript:

Specific comments:

- I would have liked to see a little more background about Fogo's eruptive history, or perhaps a reorganization of the material that is presented. For example, in the introduction, the 1951, 1995, and 2014-2015 eruptions are mentioned first, and then the "major 1680 eruption" is brought in somewhat casually. This sounds like a key event in the volcano's history, yet its importance is hard to appreciate in the current context. It's not even mentioned in the "Geologic setting and eruptive history" section. I recommend

that the entire second paragraph if the introduction (which starts with "Fogo Island features a prominent giant landslide") be integrated with the "Geologic setting and eruptive history" section. That way, the introduction covers only that material which is sufficient to understand the purpose and importance of the paper, and all of the information on past eruptions and geologic setting are contained within a specific section.

Author reply: We have integrated the second paragraph of the introduction with the second section on the "Geologic setting and eruptive history" as recommended (page 3, line 11-17; page 3, line 38 – page 4, line 6). We have also restructured the second paragraph of section 2 and added a little more information on the 1680 eruption for clarity (page 3, line 11-13). At the same time we made the attempt to not add much additional text, as suggested by reviewer 3.

- In the introduction, I found it a little odd to see that "DOWNFLOW is known to work well at steep terrain" and then in the subsequent sentence "Here we apply the model to a rather flat area." This seems contradictory. Is this one of the first studies to apply DOWNFLOW to a low-relief area? If so, is that an outcome of the paper that the model is also appropriately used where topography is subdued (assuming the model parameters can be adequately explored and constrained)?

Author reply: Indeed, DOWNFLOW has been applied mostly in steeper terrain in the past (e.g. at Mt. Etna, Mt. Cameroon, and Nyiragongo Volcano). Our study shows that the code also works well in rather flat areas (the Chã). We agree that this is an outcome of our work and not the motivation to use DOWNFLOW for the Fogo case. We have deleted the sentence from the introduction and moved it to the very beginning of our discussion about the DOWNFLOW performance (Sect. 5.3).

- There are a number of very short sections throughout the manuscript, and it might be better to eliminate or merge some of these just to keep from breaking the flow of the paper. For example, it seems to me that section 3.3.1 could be merged with section 3.3.2. It keeps the description of the DOWNFLOW simulation with no length constraint

contained within a single section, thereby lessening the chance for later confusion (in my opinion). Also, section 4.4 (the single sentence before section 4.4.1) is not really necessary and could be deleted.

Author reply: We have merged sections 3.3.1 with 3.3.2 and have deleted the single sentence of section 4.4.

- I enjoyed the discussion about the challenges of placing length constraints on the lava flow simulation. That said, I was a little surprised there was no mention of the fairly well-known work of Walker (1973) and subsequent authors, who noted the correlation between effusion rate and flow length. This could be especially important given the restricted vertical spread of historic eruptions at Fogo (making it impossible to infer a correlation between flow length and vent elevation). Given that one of the really noteworthy results of this work is a volume for the 2014-2015 lava flow, the authors could calculate discharge rates. Overall, the eruption rate (43.7 MCM over 85 days) was 6 m3/s, but it was certainly higher during the first few days/weeks of the eruption, when the flow traveled farthest. Can the authors place some constraint on the maximum effusion/discharge rate for the early part of the eruption? This might allow them to use Walker's (1973) relation between effusion rate and flow length, and could also facilitate a more thorough comparison with the 1995 eruption.

Author reply: Cappello et al. (2016, JGR) calculated a maximum discharge rate for the 2014-2015 eruption of up to 20 $m^3s$-1. We can also constrain the lava flow length using the maximum and average effusion rates for the 2014-2015 eruption and compare this to the 1995 eruption. However, when creating hazard maps we need a lava flow length constraint that is valid for a future eruption. In this case we don't have any information on effusion rates. This is the reason why we chose an empirical approach for the length constraint. That said, we will incorporate this comment and refer to the works of Walker (1973) and Cappello et al. (2016) in the final version of the manuscript.

- What is the impact of varying delta-h on the simulations? That parameter seems

relatively well constrained, but a brief mention of how important that value is to the results (what happens if the value is changed to 3.5 m? Or 2 m?) would help reinforce the simulation results.

Author reply: We have added the information that for $\Delta h < 2.5$ m and $> 4$ m, the fit significantly decreases to Sect. 3.3.1. We will also change the discussion accordingly in the final version of the manuscript.

- I was struck by the fact that Pico Pequeno, where the last two eruptions have taken place, is not the densest concentration of vents not by a long shot! This is reminiscent to me of the situation at Campi Flegrei, where Bevilacqua et al. (2015) found that the most likely location of a new vent in that caldera was not where the most recent eruption occurred. This has little direct relevance, except with underscoring the importance and challenge of knowing something about vent location when estimating hazard.

Author reply: A valuable comment. We will discuss the challenge of knowing future vent locations and emphasize the importance of such knowledge for hazard assessments in the discussion of the final version of the manuscript.

- I really like the coherence maps showing lava flow area, but wonder if those might be summarized in a single figure? The maps dominate the figures (in terms of their size and the space they take up), even though they are not necessarily the most critical figures in the paper. One option might be to move the coherence images to the appendix, and replace the separate panels of Figure 7 with a single map that shows the flow outline at different times (because there is no look angle artifact with coherence, different dates could be combined on a single map). This would make it easier to compare what are now separate panels and show the spatio-temporal growth of the flow at a glance. A disadvantage is that one would not necessarily "see" the small flow that was active in mid-January, since it is mostly contained within the existing flow. That's why the coherence images could remain in the appendix, but the areal growth of the flow could be represented in a single manuscript figure that would communicate that

information more succinctly. Also, why do the coherence images not go through the end of the eruption, in early February? If they show no change in the flow (aside from incoherence due to cooling and rapid subsidence), that would be helpful in constraining the effusion rate (i.e., that there was little volume erupted during late January – early February). This is already hinted at in section 4.2, but there is little evidence given to back up claim of a post-January 16 erupted volume of 0.05 MCM.

Author reply: We have replaced the separate panels of Figure 7 with a single map that shows the flow outline at different times. We have also adjusted the text accordingly (Sect. 4.1). This way we lost the actual coherence information. For the reader to be able to refer to our coherence interpretation, we will move the coherence images to Appendix D in the final version of the manuscript. We will also add coherence information for the very end of the eruption (19 January – 10 February 2015) there.

- The manuscript does a good job of pointing out that topography must be updated in order for subsequent DOWNFLOW runs to be relevant, and indeed, this is borne out by the results some areas had a 0% chance of inundation based on the pre-eruption DEM, but as the 2014-2015 eruption progressed, the modified topography caused these areas to become inundated. I think this could be stated more explicitly, however. Lava flows make their own topography, so the failure of the simulation to exactly predict the flow coverage (the "0%" probability areas) is really more a reflection of the changing topographic conditions than of the capability of the simulation, right? This is a major conclusion of the paper, but it is somewhat spread out between sections 4 and 5. The authors might want to make it more explicit. Perhaps even address the question about iteratively updating topography (with TLS?) during a crisis so that successive DOWNFLOW runs will be more representative of changing conditions. That seems like an unstated conclusion, given the datasets the authors used and the simulations they ran.

Author reply: We will emphasize this point more in our conclusion in the final version of the manuscript.

- Was any adjustment applied to the volume calculation to account for vesicularity? For aa flows, 25% is often assumed. If no adjustment is made, the authors should be explicit that the volume is a bulk volume, and that vesicularity is not accounted for. This might explain a small percentage of the difference with the Ferrucci volume

Author reply: In the discussion (Sect. 5.2) we have added the information that the DEM difference method gives a bulk volume while the Ferrucci et al. (2015) and Cappello et al. (2016) both calculate DRE lava flow volumes.

Technical recommendations:

- I suggest deleting from the abstract the sentence "Based on this, we discuss how our study can help improving the general understanding of basaltic lava flow behavior. " Such a discussion really doesn't exist in the manuscript, and I don't think anything is lost from the abstract if this sentence is removed (in fact, the abstract becomes tighter).

Author reply: We have deleted the sentence "Based on this, we discuss how our study can help improving the general understanding of basaltic lava flow behavior." from the abstract.

- The authors reference "Harris and Rowland, 2001" in the introduction section to highlight their FLOWGO code, but maybe referencing their 2015 update to that code would be better? That paper is included in the AGU monograph on Hawaiian volcanism.

Author reply: We have changed the reference to the updated version of the FLOWGO code (page 2, line 13).

- Toward the end if the introduction, it is unclear whether "featuring a 5 m spatial resolution" is referring to the pre-eruption DEM, the post-eruption DEM, or both.

Author reply: The 5 m spatial resolution is referring to both DEMs. We have made this clear by adding "both" to this sentence (page 2, line 35).

- Is the last paragraph of the introduction ("In the first section of this paper") necessary?

[Figure]

I think the introduction might be more powerful if it were to end with mention that this work is the first to use TLS data as a base to develop probabilistic hazard maps.

Author reply: We have deleted the last paragraph of the introduction.

- In the first paragraph of section 2, note that Bordeira reaches a height (not an elevation) of 1,000 m above the Cha.

Author reply: We have corrected this.

- The first sentence of the second paragraph of section 2 is awkward, and should be rephrased. It implies that reports of eruptive activity exist for the period 1500-1660, but afterwards there was less information? This lesser-known period would include the time period of the "major 1680 eruption."

Author reply: We have reworded the paragraph for clarity.

- At the end of section 2, I suggest replacing the word "reenacted" with "renewed."

Author reply: We have corrected this.

- In section 3.1, the phrase "ranges between 12 h (for the ascending and descending pairs 57/64 and 148/155) to 6 days" is awkward and should be reworded. I think the authors mean that consecutive ascending and descending data are offset by 12 hours, but the two sets of A/D pairs are offset by about 6 days. In any case, this might be confusing to readers who don't regularly work with TSX data. Also in this section, it might be useful to mention that vegetation is not an issue in the Cha, so coherence really just reflects steep slopes and surface change. This is stated much later in the manuscript, but should probably also be explained here.

Author reply: We have deleted the sentence about the time delay between the orbital paths of the satellite to avoid confusion and we have added a sentence on the main decorrelation factors in the study area at hand.

- In section 3.2.2, I was a little confused by the scanner locations and positions. It

appears that there were three major locations from which scans were collected Beco, Saia, and Amarelo and at two of these locations, multiple positions were occupied (presumably with different fields of view). Is that right? After the three locations are mentioned, it is stated that "At five of the scanner positions" GPS data were acquired. It is not clear if these 5 positions are distributed between the Beco and Amarelo sites, or represent all of the positions at these sites, etc. I recommend that a little rewording be done here to make the procedure easier to understand.

Author reply: Indeed, there are three major scanner locations from which scans were collected (Beco, Saia, and Amarelo). At Beco and Portela, multiple positions were occupied with different fields of view. We are referring to Table 1 when stating that we collected GPS data at five of the scanner positions. In Table 1 we show for which scans we collected GPS data. We have now added the information that GPS data was collected for one position on Monte Saia and for four positions on Monte Beco to the text (Sect. 3.2.2, page 5, line 24-25). However, we will change the text in the final version of the manuscript as suggested to make the whole procedure easier to understand.

- Toward the end of section 3.2.2, the phrase "here used methodology" is awkward and should be reworded ("methodology used here" would be fine).

Author reply: We have corrected this.

- In section 3.4, I would delete mention of the "82,000 vents," since it only raises the question of how that number was determined. Since this is explained in section 3.4.3 in greater detail, the earlier mention can be removed.

Author reply: We have removed the mention of the 82,000 vents in Sect. 3.4.

- In section 3.4.2, the phrase "which integrates up to be 1" should probably be reworded to something like "which sums to 1."

Author reply: The integral of the PDF over the entire domain is 1. The pdf does not

sum to 1. We decided to delete the second part of the sentence as it does not add anything but confusion.

- In section 3.4.3, right after equation 2, "extents" should be changed to "extends."

Author reply: We have corrected this.

- In section 4.1, did the flow thicken after it stopped advancing (after December 23, 2014)? From figure C, it looks like it did (at least, the active lobe in January did), and that would probably be worth stating directly.

Author reply: We now mention in Sect. 4.1 that we assume the flow thickened after it stopped advancing, as this is what we observe from the active lava flow in January (Appendix C).

- In the first paragraph of section 4.2, I didn't understand what was meant by the error being smaller "when comparing post-eruptive and pre-eruptive grids".

Author reply: We firstly compare the point cloud to the pre-existing DEM. Then we compute a 5 m grid from the point cloud and compare this to the pre-existing DEM, which gives a slightly smaller RMSE. We have reworded the sentence to make this clear.

- In the second paragraph of section 4.2, isn't the area calculated from the coherence maps, and not the topographic difference? Also, note that the area given here (4.84 km2) differs from the area given at the end of section 4.1 (4.85 km2). Finally, perhaps the maximum thickness of the 2014-2015 flow could be given along with the average thickness, instead of at the end of the section?

Author reply: The area is indeed calculated from the coherence maps. We actually also used the DEM difference map to check the area and the value was slightly smaller (as we are missing data at the westernmost tip of the northern lava lobe). We have therefore deleted the area from Sect. 4.2. We have also restructured the paragraph to first talk about the thickness (including the maximum thickness) and then about the

lava flow volume.

- In section 4.3, I thought it was a little awkward to bring up the apparent correlation between the simulation and the thickness, since it is not raised again until well into the discussion. Maybe wait until the discussion to note this similarity? That way, it doesn't get in the way of the description of the simulation results.

Author reply: We have deleted the sentence "We find intriguing similarities between the simulation and the lava flow thickness in several areas, including the initial flow (through point #1 in Fig. 8a), as well as the NW (#2), W (#3), and S (#4) lava lobes" from Sect. 4.3. We now only mention this correlation once in the discussion (Sect. 5.3).

- The first time a percentage is given in terms of a DOWNFLOW result (in section 4.4.1), it might be useful for the authors to offer a brief explanation of what that percentage is referring to for example, is it the likelihood that a future eruption will inundate a specific pixel? Just so that the reader is clear on the meaning.

Author reply: We have added the sentence "Our hazard maps show the probability of lava flow invasion, i.e. the likelihood that a future lava flow will inundate a specific pixel." to the beginning of Sect. 4.4.1.

- Toward the end of section 4.4.2, "remarkable hazard" is an awkward phrasing that should be reworded. It's unclear if 10% is remarkable because it is so low or so high.

Author reply: We have reworded this. We consider a lava flow hazard of 10 % rather high (referring to the "Fogo scenario").

- The idea of "catchment" maps is a good one. The authors may wish to reference some work along the same lines at HVO by Frank Trusdell and Jim Kauahikaua, who use that technique for hazard assessment on the Island of Hawaii.

Author reply: We will add the references mentioned here.

- At the end of section 5.2, is there any indication why the volume derived from the topographic difference is so much greater than that of Ferrucci et al., 2015? The idea that volumes inferred from thermal data might be so much different from those determined by topographic differences is a little unnerving.

Author reply: Ferrucci et al., (2015) and Cappello et al., (2016) estimate vesicle-free volumes, while we refer to the bulk volume (estimated using the DEM difference method). We have added this information to the discussion and also refer to the often assumed vesicularity of 25 % for aa flows. However, this doesn't fully explain the different values. Future studies are needed to address this issue.

- In the third paragraph of section 5.4, the authors raise the question of why the 1995 and 2014-2015 flows followed such similar paths. But isn't the answer "topography"? Can it be anything else? It's unclear to me what type of "future studies" might actually address this question.

Author reply: We are here talking about "the magma", referring to the (subsurface) dike propagation, rather than the lava flow paths on the surface. We have done some rewording to avoid misunderstandings.

- In the conclusions section, I would recommend deleting the second-to-last sentence. "We conclude that the next lava flow will very likely change the lava flow hazard within the Cha again." This sounds rather grand, but is also pretty plain for all to see, and the point is made more effectively earlier.

Author reply: We have deleted the sentence "We conclude that the next lava flow will very likely change the lava flow hazard within the Cha again."

Please also note the supplement to this comment:
http://www.nat-hazards-earth-syst-sci-discuss.net/nhess-2016-81/nhess-2016-81-AC1-supplement.pdf

[Figure]

[Figure]

**Supplement:**

[revised manuscript text omitted]
, 2014), about 27 eruptions occurred at Fogo Volcano (Ribeiro, 1960; Day et al., 2000). In 1680 a major eruption was associated with strong earthquakes and caused people to flee the island temporarily, according to historic records (Ribeiro, 1960; Day et al., 2000). The 1785, 1799, 1816, 1847, 1852 and 1857 eruptions produced lava flows travelling seawards on the eastern flank of the volcano. In the 1860s the first settlements were established within the Chã, likely because of the abundance of fertile volcanic soils. However, the last three eruptions (which occurred in 1951, 1995 and 2014-2015) took place within the Chã, where villages and agricultural lands were affected (GVP, 2013; Torres et al., 1997 in Texier-Teixeira et al., 2014).

On 2 April 1995 a fissure eruption started at Pico Pequeno (Amelung and Day, 2002), a small cone WSW of Pico do Fogo (Fig. 1). All residents were evacuated, but houses as well as ~4.3 km$^2$ of agricultural land were destroyed (GVP, 1995a). The total area covered by lava flows during this eruption was estimated to be around 4.7 km$^2$ (Amelung and Day, 2002). The flow thickness ranged between 1 m and ~20 m (GVP, 1995b; Worsley, 2015). After a period of increased strombolian activity, the eruption ended on 26 May 1995. Because some of the best farmland on the island is located within the Chã, people moved back after the eruption despite attempts from officials to relocate the population.

Almost 20 years later, on 23 November 2014, a new eruption started at Pico Pequeno with the opening of a fissure located ~200 m southeast and roughly in parallel to the 1995 fissure (González et al., 2015). At first, six active vents were emitting lava fountains, ash and gas. Later on, activity focused at one major vent (located at 24.35341° W and 14.9446° N). Lava flows were emitted from the base of Pico Pequeno and travelled to the southwest before splitting into two main lobes that we will refer to as the northwest (NW) and south (S) lava lobes throughout this paper. The main road and evacuation route out of the Chã was already cut off two days after the eruption had started and the residents of the Chã were forced to transport their movable property uphill before being evacuated from the area (Fig. 2a). The NW lava flow continued to engulf Portela and Bangaeira in early December 2014 and then gradually ceased (Fig. 2b). However, a new lava lobe split from the NW flow closer to the eruption site and advanced westward towards the Bordeira wall (referred to as W lobe throughout this paper). This lava lobe split into a northern and a southern lobe close to the Bordeira wall and covered the houses of the small agricultural settlement of Ilhéu de Losna. In early 2015, effusive activity was replaced by increased strombolian explosions at the vent (cf. Appendix A) before the eruption ended on 8 February 2015 (GVP, 2014). Shortly after the end of the 2014-2015 eruption, reconstruction of buildings and infrastructure within the Chã had started again (Fig. 2c), despite renewed efforts from officials to suppress permanent residence within the Chã.

After the 1995 Fogo eruption, lava flow simulations were tested on the base of a 15 m Digital Terrain Model (DTM) (Quental et al., 2003). The Cellular Automata (CA) technique was applied to simulate the time and space dependent flow emplacement. Results were in agreement with the actual lava flow coverage, but only the first two days of the 1995 eruption were reproduced successfully. A general, yet provisional, volcanic hazard map for the scenario of a renewed phreatomagmatic explosive eruption comparable to the major 1680 eruption was provided by Jenkins et al. (2014) on the base of investigations by Day and Faria (2009, unpublished). This map suggested that the entire area of the Chã as well as

the eastern flank of the volcano were areas of high hazard resulting from lava flows, 2-10 m ash fall, possible pyroclastic surges and rock avalanches. Furthermore, Cappello et al. (2016) use HOTSAT satellite data and the physically-based MAGFLOW model to simulate the lava flow emplacement in rapid response during the ongoing 2014-2015 eruption. In the study at hand, we constructed probabilistic lava flow hazard maps for the Chã das Caldeiras and the eastern flank of Fogo Volcano. We were particularly interested in whether the 2014-2015 eruption has significantly changed the lava flow hazard in the affected areas, as this has important implications in temporal and spatial changes of lava flow hazards in general.

**3 Data and Methods**

One of the first needs immediately upon an effusive crisis is an up-to-date model of the new topography. In rapid response to the 2014-2015 Fogo eruption, a Hazard and Risk Team (HART) of the German Research Centre for Geosciences (GFZ) processed high resolution satellite radar data and went to Fogo Island in order to acquire high resolution topographic data between 11 and 21 January 2015. The topographic data are needed to estimate lava flow characteristics, such as erupted volumes, and serves as the most crucial input data for our lava flow simulations and hazard assessment. The frequent Synthetic Aperture Radar (SAR) satellite data acquisitions allow us to map lava flow emplacement over time, information that help us to better understand the lava flow model performance.

**3.1 SAR data**

We used SAR data acquired by the German satellite TerraSAR-X (TSX) to monitor the emplacement of the 2014-2015 lava flow. The TSX satellite operates at a wavelength of 3.1 cm (X-band) of the electromagnetic spectrum. The data were acquired in the satellite's Spotlight mode (~1 m spatial resolution, scene size ~10 km × ~10 km), horizontal polarization, and have a repeat time of 11 days. TSX data are acquired over Fogo Island on four tracks, two ascending (orbital paths 57, incidence angle 53.5° and 148, incidence angle 38.9°) and two descending (orbital paths 64, incidence angle 27.9° and 155, incidence angle 46.3°). We use coherence measurements to monitor lava flow emplacement every 6 days between 14 November 2014 and 28 December 2014 (i.e. the early, highly effusive phase of the eruption) and every 11 days thereafter until 10 February 2015.

Interferometric coherence is a measure of the correlation between the phase components of two SAR images of the same track (i.e. the same viewing geometry) (Hanssen, 2001). Coherence values range from 0 (low coherence, decorrelation) to 1 (high coherence, strong correlation between SAR acquisitions). As a consequence of the time delay between two acquisitions (11 days for TSX), temporal decorrelation occurs in repeat-pass InSAR as the scatterers within a resolution cell move, change their dielectric properties or are replaced by a new set of scatterers (e.g. upon lava flow emplacement or ash deposition) (e.g. Zebker et al., 1996). In the study area at hand, decorrelation is mostly associated with steep slopes and surface change, as little vegetation coverage is found within the Chã.

[revised manuscript text omitted]
 7 shows the areal coverage of the 2014-2015 lava flow over time as mapped using TerraSAR-X coherence images. This map shows that the NW lava lobe had already traveled almost 4 km within the first two days of the eruption, but had not quite reached the village of Portela yet. The S lava flow was also already emplaced. This lava lobe did not advance much after emplacement, except for some minor widening. Between 20 November 2014 and 1 December 2014, primarily the NW lava flow widened, destroying the first houses of Portela. We also observe minor propagation and widening at the W lava lobe. This trend of widening and engulfing more houses of Portela continued throughout 6 December 2014. Until that time, lava flowed in a well-defined channel north of Monte Saia (cf. Fig. 8a, point #1 and Fig. 8b, profile C-C'). According to our coherence analysis, most of Portela and Bangaeira were covered by lava flows in the period between 6 and 12 December 2014. Until 17 December 2014 the W lava lobe was propagating, but the settlement of Ilhéu de Losna was not yet harmed. Between 17 and 23 December 2014, this lava flow advanced further towards the west, where it split into two N and S sub-lobes after reaching the Bordeira wall. At this point the third, smaller settlement of Ilhéu de Losna was destroyed. Since that time until the end of the eruption, the lava flow continued to thicken (e.g. we observe thickening of the active lava flow in

January 2015, see Appendix C) but the lava flow extent stabilized, implying that effusive activity had slowed down. Only minor widening was observed at the W lava lobe between 23 and 28 December 2014. Also, during fieldwork, we observed an active surface flow on 12 January 2015 and minor propagation of the lava flow in-between Monte Saia and Monte Beco. Details on the active surface flow are provided in Appendix C. Even though the eruption lasted until 8 February 2015, no lava flows were active in the last period of the eruption. The final boundary of the 2014-2015 lava flow as shown in black in Fig. 7 encloses an area of 4.85 km². We provide the TerraSAR-X coherence maps in Appendix D. Therein 
[revised manuscript text omitted]
 a total erupted bulk volume of the 2014-2015 lava flow of $43.7 \times 10^6$ m$^3$ +/- $5.2 \times 10^6$ m$^3$; assuming a 25 % vesicularity, as often used in literature (e.g. Wolfe et al., 1987; Poland et al., 2014), the dense-rock equivalent (DRE) value is ~ $33 \times 10^6$ m$^3$. Our DRE volume estimate is three times larger than previous estimations by Ferrucci et al. (2015). Cappello et al. (2016) calculated a total, DRE lava flow volume of $15 \times 10^6$ m$^3$ from time-averaged discharge rates as inferred from HOTSAT satellite data which, according to our results, is underestimating the 2014-2015 lava flow volume. The reasons for this disagreement need to be addressed in future studies as lava vesicularity alone cannot fully explain this difference. According to our results, the 2014-2015 lava flow has a very similar surface coverage (4.85 km$^2$) and volume ($43.7 \times 10^6$ m$^3$ +/- $5.2 \times 10^6$ m$^3$) to the lava flow of the 1995 eruption (area: ~4.7 m$^2$, volume: ~$46 \times 10^6$ m$^3$) (Amelung and Day, 2002).

**5.3 DOWNFLOW performance**

The Fogo case study represents a successful application of the DOWNFLOW algorithm. DOWNFLOW is known to work well at steep terrain (Favalli et al., 2009a, 2011b; Tarquini and Favalli, 2011). In this study the DOWNFLOW simulation has proved to perform well on rather flat areas, like the Chã das Caldeiras. This implies that lava flow paths are largely controlled by the topography even, and maybe especially, in relatively flat terrain. Furthermore, our Fogo example demonstrates the first application of DOWNFLOW to a TLS dataset. In order to discuss the DOWNFLOW performance, we compare the simulation (Fig. 8a) to the real lava flow coverage (Fig. 8b and Fig. 7). The very early phase of the 2014-2015 eruption, when lava travelled in a well-defined channel according to the TerraSAR-X coherence analysis (Fig. 7a-c) and the thickness map (Fig. 8b and profile C-C'), is reproduced by the DOWNFLOW simulation in great accuracy (Fig. 8a). Furthermore, it seems that lava flows are thicker where the number of paths crossing a pixel by the simulation, are highest (e.g. #2, #3, #4, and across profile C-C' in Fig. 8b). This applies especially to topographic ponds, where the filling algorithm, which is implemented in the DOWNFLOW model, causes the lava flow simulation to fill in the local topography before continuing the path downslope (profiles A – A' and B – A'). This also explains the fact that, even though the lava flow had reached the first houses of Portela within the first days of the eruption, it stopped for a couple of days, while ponding, before continuing its path downslope, overflowing the villages of Portela and Bangaeira (Fig. 7, Sect. 2 and Sect. 4.1). We conclude that, both channeling and ponding can be very well simulated by the code.

Differences between the simulation and the real lava flow coverage occur due to two main facts: first, the DOWNFLOW simulation runs until the lava flows hit the end of the DEM, while the actual lava flows stop when effusive activity ceases. Second, the DOWNFLOW simulation starts only once at the vent location and then keeps running downslope, while the real lava flow is produced iteratively depending on the supply rate at the vent. This process is creating new topography upon emplacement, which changes the paths of lava emplaced during subsequent effusive pulses. This observation explains why the DOWNFLOW simulation fits the extent of the lava flow during the first few days of the eruption almost perfectly in close proximity to the vent while the distant 2014-2015 lava flow fronts are in good agreement with topographic ponds and therefore a higher number of times that a pixel is hit by a simulation.

With these findings, the 2014-2015 lava flows of Fogo Volcano provide intriguing examples of the impact of local geologic structures, such as topographic channels and ponds, on lava flow pathways and the ability of numerical lava flow simulations to reconstruct and predict these. We suggest that updating the local topography, even during an ongoing eruption, is of importance in order to forecast paths of lava flows produced by subsequent effusive pulses.

**5.4 Lava flow hazard maps**

Our volcano-wide hazard maps (Fig. 10a and 10b) allow speculations about infilling mechanisms of giant landslide amphitheaters of volcanic origin. We find that high lava flow hazard areas are located mainly along the wall of the landslide scarp. Flows are then likely to follow pathways down the flanks, along the edges of the scarp. We would expect generally similar main lava flow hazard patterns for other ocean islands with infilling landslide amphitheaters and comparable topographic structure, such as Piton de la Fournaise (La Réunion, France) or Teide Volcano (Tenerife, Spain).

However, regarding the lava flow hazard estimation, we are left with uncertainties. The DOWNFLOW hazard map generation depends on the topography, $\Delta h$, the PDF of vent opening, and the lava flow length constraint. With our new post-eruptive DEM, we have a very reliable dataset for the DOWNFLOW simulation, at a resolution that meets, or even exceeds the requirements of the model. The parameter $\Delta h$ has a wide range of fit according to the calibration (Sect. 3.3.2, Fig. 4), meaning that the parameter $\Delta h$ is not among the main sources of error. As for the PDF, previous studies have shown that even with a low number of vents, the resulting vent distribution is robust (Tarquini and Favalli, 2013). In contrast, the lava flow lengths constraint is known as a potential source for large errors (Tarquini and Favalli, 2013). In our Fogo case study the historic record is sparse, which causes the lava flow length constraint to be poorly defined. Overestimating the lava flow length would produce hazard maps with high hazard zones smeared, or extended downhill. Underestimating the lava flow length produces hazard maps with high hazard zones that shrunk uphill. To minimize the introduced error, we take into account a rather large range of possible flow lengths.

Future studies are needed to address the question why very similar subsurface pathways were reused and adjacent dikes developed during the 1995 and 2014-2015 eruptions (Amelung and Day, 2002; González et al., 2015). 
[revised manuscript text omitted]
. FCT also funded the GPS data processing in the framework of the FIRE (PTDC/GEO-GEO/1123/2014) project. We thank Eleonora Rivalta, Jacqueline Salzer and Adam Mehlhorn for valuable suggestions which helped improving the manuscript. We are very thankful for detailed and constructive comments from Michael Poland, Matthieu Kervyn, and Sónia Calvari which greatly improved the manuscript. We also thank Paulo Fernandes Teixeira and Lourenco Francisco Fernandes for their invaluable assistance during fieldwork on Fogo Island.

**References**

[revised manuscript text omitted]

---

## Author Comment (AC2) · 19 May 2016

We very much appreciate the provided comments. We incorporated all suggestions into the manuscript as follows (please refer to the AC1 supplement for an updated, but still intermediate version of the manuscript):

Page 1, line 35

Author reply: Typo corrected

Page 2, line 8: reference missing for "flank failure in the Quaternary period"

Author reply: Comment accepted, however, we have moved this paragraph to section 2 and deleted this sentence in response to a comment from reviewer 1.

Page 2, line 11/12: reference missing for "Since that time three major eruptions oc-

curred at Fogo Volcano (1951, 1995, and 2014-2015), all of which affected the Chã."

Author reply: Reference added. We have moved the whole paragraph to section 2 in response to a comment from reviewer 1 (page 3, line 11-17).

Page 3, line 7: TLS

Author reply: TLS explained (now page 2, line 32).

Page 3, line 22/23: reference missing for "Fogo Island is one of the youngest volcanic islands of the Cape Verde Archipelago in the Atlantic Ocean and is built up from the remnants of one single giant volcano, known as the Monte Amarelo Volcano."

Author reply: Reference added (page 3, line 1-2).

Page 4, line 17 GFZ

Author reply: GFZ explained (page 4, line 2).

Page 4, line 18: 11 January 2015 and 21 January 2015

Author reply: First case of January 2015 deleted (page 4, line 11).

Page 4, line 26: put per instead of x here: "scene size ~10 km x ~10 km"

Author reply: We have changed the "x" for the mathematical version " $\times$" (page 4, line 18).

Page 4, line 29: put and instead of to in the sentence: "The temporal offset between the tracks ranges between ~12 h (for the ascending and descending pairs 57/64 and 148/155) to ~6 days."

Author reply: We have deleted this sentence according to the suggestion of reviewer 1.

Page 9, line 18 and page 9 line 28: different values for lava flow coverage

Author reply: We took the value from the coherence mapping (page 9, line 6).

Page 9, line 30, 31, 34, 38 and 39(2x): per instead of x

Author reply: We have changed the "x" for the mathematical version " ×" (page 9, sect. 4.2, paragraph 3)

---

## Author Comment (AC3) · 31 May 2016

Author reply: We thank the reviewer for very detailed and constructive comments and we will revise the manuscript accordingly.

Major comments:

1. My main concern is that the authors stress throughout the manuscript the relevance of their study for risk management, the rapid acquisition strategy of their method and the role of the HART of GFZ in assisting local decision makers during the crisis management and in training local residents. Although I do not doubt that the high quality products presented might be relevant for crisis and long term risk management, it is unclear in the manuscript to which extent the local actors were indeed informed about these results and how much these maps were used to inform the local population.

[Figure]

Either the authors have taken actions to ensure that the scientific results have a direct impact on risk management and information to local population, and they should describe it, or they have not (yet) done so and assume that a scientific publication is sufficient to have an impact. In the latter case, the argument of the impact of the research for risk management should be downscaled in the paper (eg. page 2, lines 1-6, page 12 lines 5-10, page 15 lines 37-38). As pointed out in the introduction, residents tend to re-occupy zones invaded by lava flows but from experience I don't believe accurate lava flow hazard maps can make a difference without major investment in education and communication actions. This is mentioned by authors (page 2, line 6) but this should be clear in the discussion and conclusion.

Author reply: This is a valid point. We do agree with this comment and we will downscale the relevance of this study for risk management in the revised manuscript. In the revised version, we focus on hazard assessment and provide comprehensive hazard maps. Neither vulnerability nor preparedness have been investigated by us. However, our hazard maps might be valuable for risk management, provided that they are scientifically peer reviewed and subsequently communicated to the local decision makers. We took two important steps to assure that our results are accessible. Firstly, we chose to publish in an open access journal. Secondly, we collaborated with Sonia Silva (who is an author on the manuscript), the leading local volcano scientist. She interacts with the local decision makers and will be able to use our results and communicate them further.

2. DOWNFLOW: I am quite familiar with the approach and capabilities of the DOWNFLOW code. In some places the authors should be more careful in the description of the DOWNFLOW results and be critical. On p 10 (lines 4-10), authors highlight the good match between the DOWNFLOW simulation and the outline of the actual flow. Where this is true for some areas (points 1, 2, 3, 4 on Fig. 8), this is not so true for zone 5 where the probabilities of DOWNFLOW are much lower than other zones located at shorter distance (Western and Southern border of the calder). Zone of overestimation of the DOWNFLOW simulations should also be described in this part of the results. Also in the discussion (section 5.3), authors should not only highlight the capabilities but also the limitations: between the Northern and NW branches, a lot of pixels have a low probability of lava flow invasion but were not invaded, whereas the opposite is true for Zone 5. These uncertainities, and their cause, should be highlighted. When discussing accuracy (page 13, line 31), quantitative values should be provided: the reader should be informed that 'very good' simulation have accuracy parameter of ~0.5 even without considering the issue of length.

Author reply: We agree this is a valid comment and we will include changes following from this remark in the final version of the manuscript. We will highlight uncertainties and give quantitative values for accuracies.

Minor comments:

- Page 3, line 7: spell out TLS when first used in main text

Author reply: We accept this comment and changed the text accordingly (we now spell out TLS on page 3, line 7).

- A recently published paper by Cappello et al. (2016, JGR, DOI: 10.1002/2015JB012666) also discuss lava flow modelling for the Fogo 2014-15 eruption. As that publication use a physically-based model, a comparison of the advantage and limitation of the two approaches in the discussion would be useful.

Author reply: We accept this comment and incorporated changes accordingly. While the purpose of Cappello et al. (2016) is near real time lava flow hazard estimation during the 2014-2015 eruption, our ultimate purpose is to create hazard maps that are valid for the next eruption of Fogo Volcano. We include this very recent study in the revised manuscript and compare the different approaches in our discussion.

- Paper by Albino et al. (JGR, 2015) presents a volume estimate for the Nyamulagira 2011-12 lava flow eruption using a TanDEM-X DEM. The author could compare the

accuracy of their DEM comparison and volume estimate with the one presented by these other authors. Why was the TanDEM-X technology not applied in the case of the Fogo eruption to derive the post-eruption DEM?

Author reply: We accept this comment and incorporated changes accordingly. When using the DEM difference method for lava flow volume calculations, errors in volume only occur due to the quality and the resolution of the DEMs. Albino et al. (2015) used TanDEM-X DEMs to estimate the volume of the lava flow of the 2011-2012 Nyamulagira eruption to be 305.2 +/- 36.0 x 106 m3. Their error therefore corresponds to 11.8 % of the total lava flow volume. Our error corresponds to 11.9 % of the total lava flow volume. The achieved DEM qualities are therefore comparable, which we discuss in the revised version of the manuscript. We still have to note that our pre-eruptive DEM is of lower resolution and quality as compared to our post-eruptive DEM. We used ground-based techniques, because TanDEM-X bistatic data is not available for Fogo after the 2014-2015 eruption.

- The authors argue that ground-based technology are 'more flexible' (page 3, line 5): I find this argument a bit weak, as ground-based technology require to access inhospitable volcanic area during or directly after an eruption. This is also contradicted by the discussion where the TLS approach is described as 'time consuming and challenging' (page 13, line 10).

Author reply: We refer to the sentence before, satellite data need to be tasked and ground-based methods are more flexible with respect to their acquisition time and date. For instance, we produced the first post-eruptive DEM from ground-based data in January 2015, while the next (and only other) available post-eruptive DEM data was acquired on 20 June 2015 by the Pléiades satellite, more than 5 months after the end of the 2014-2015 eruption. In section 5.2 we compare ground-based TLS to ground-based SfM techniques and conclude that TLS in this comparison is more time consuming and logistically challenging. We have done some rewording to avoid misunderstandings.

- HART: authors repeatedly highlight the action of GFZ-funded HART initiative (page 4, line 15-18; page 12 line 3-10) and the rapid-response character of the action (page 4, line 16). Although HART is for sure a nice initiative I don't think this paper should aim at giving so much publicity for it. The author should also justify why they consider that the topographic survey is part of a 'rapid response': as eruptions are often separate by several years, this survey could be done once it is clear that the eruption is finished and access to the site is secured.

Author reply: We accept this comment and changed the text accordingly (i.e. we reduced the publicity for the HART initiative).

- Section 3.2.4: mention already here the spatial resolution of the post-eruption DEM produced

Author reply: We included the spatial resolution of the post-eruptive DEM in this section (in the sentence: "We generate a DEM featuring a 5 m spatial resolution from more than 164 million TLS data points and mosaic it with the pre-eruptive DEM").

- 'filling algorithm': in the methodology (page 6, line 33; page 10, line 10), as well as the discussion section (page 13, line 33), the authors refer to a 'filling algorithm' integrated in DOWNFLOW. More information should be given about this aspect. DOWNFLOW being a probabilistic model with no explicit lava flow thickness and topography adaptation, I don't really understand what is meant by 'filling algorithm' except for the possibility of the simulation to continue beyond actual pits in the DEM. How this is implemented in the algorithm should be explained in details. The observation that higher probabilities are found in pits, corresponding to thicker flow accumulation, although interesting, is not a surprise as it is a simple results of the topography-control on the lava flow paths. I would not say that these similarities are 'intriguing' (page 10, line 5), they are rather expected and logical based on the modelling approach.

Author reply: We added more detail on the filling algorithm to the text. We also added relevant references. The "filling algorithm" simply adds (deposits) a certain thickness

when there is a local minimum. A local minimum is defined as follows: During the simulation of one steepest decent path, the topography is randomly perturbed (within the interval $\Delta h$). And if the simulation is in a minimum, the topography is perturbed again. If for 10 times the simulation is still stuck in a minimum (i.e. the simulation doesn't find a steepest decent path to follow downslope) a certain thickness is added. (Of course the number 10 can be changed, but we think 10 is not too little (not to introduce a local minimum just due to the perturbation) and not too big (because it will use CPU in these evaluations). Typically a 'very' small thickness is added (in our case 0.01 m) during iterations. This value can be adjusted in the DOWNFLOW input file. The smaller the value the longer it will take for the simulation to run. But the smaller the value, the better is the filling. According to a suggestion from reviewer 1, we do not longer mention the comparison of the thickness and the simulation in section 4.3. We only mention this in the discussion now.

- Section 3.3.2: the optimal $\Delta h$ value is defined based on the maximization of the best fit parameters, using the actual lava flow as reference point. In the discussion, authors argue that this parameter has a wide range of fit (page 14, line 19), although figure 4 actually suggests that for $\Delta h$ <2.5 m and >4 m, the fit significantly reduces. How confident are the authors that this $\Delta h$ value will also be optimal for the future eruption?

Author reply: We agree and we have added the information that for $\Delta h$ < 2.5 m and > 4 m, the fit significantly decreases to Sect. 3.3.1 (Sect. 3.3.1 and Sect. 3.3.2 are now merged). We will also change the discussion according to this comment in the final version of the manuscript. The $\Delta h$ value represents the characteristics of the lava. According to previous applications, this parameter differs for different volcanoes, but not so much between different eruptions of one volcano. We will mention this and add references to the discussion of the final version of the manuscript.

- Section 3.4.1.: Authors should clarify that they assume a linear decline in probability from the minimum to the maximum length, similarly to previous application of DOWN-FLOW. Bonne et al. (Int. J. Remote Sensing) demonstrated that for Mt Cameroon, a

Gaussian probability decrease better fitted observed lava flows' lengths.

Author reply: We accept this comment and incorporated this into the manuscript.

- Page 8, line 2: justify the bandwidth of the Gaussian kernel. This bandwidth can have a major impact on the resulting PDF map.

Author reply: This is a valid point. We discuss this in the revised version of the manuscript and we included a study published by Bartolini et al. (2013) who suggest how to calculate an optimal bandwidth. According to an equation from that paper (Bartolini et al., 2013) the optimal bandwidth for our case study would be 3600 m, i.e. we would have only one maximum in our distribution, centered on the cone itself. However, we think that the cone is clearly not the maximum of probability of vent opening considering recent eruptions. Therefore, we chose the value by trials, not to have the distribution too undersmoothed or too oversmoothed.

- Equation 2: explain why the resolution of the DEM $\Delta x$ and $\Delta y$ need to be taken into account in this equation.

Author reply: In equation (2) we have a product of probabilities (the uppercase P are probabilities, which we state more clearly in the revised text). The probability of vent opening in a pixel is rhoVj $\Delta x$ $\Delta y$. This is because rhoVj is not a probability, but is a PDF (probability density function), that is a probability divided by an area. So the probability of having a vent in a given pixel is rhoVj $\Delta x$ $\Delta y$. We will include this in the final version of the manuscript.

- Fig. 8a: why is the color bar of this figure not presented in a quantitative way (with percentage) similarly to Fig. 9 and 10. This would be much more interesting. Actually the color scale of Fig. 10 is the most interesting one, as it also enable to know what is used as lower threshold (pixel with no color).

Author reply: Fig. 8a shows the DOWNFLOW simulation output, i.e. a grid with the same cell size as the input DEM, where each pixel value gives the number of n steepest

decent paths overlaying the cell area. The highest n values reflect the most likely paths followed by the lava downhill. In such a single DOWNFLOW simulation (with N number of runs), the lava flow length is not constrained (all steepest decent paths run until they reach the end of the dataset) and therefore this map does not reflect a "lava flow hazard". To avoid confusion with a hazard map, we chose to color scale the map according to a "high" or "low" number of times that a pixel is crossed over the total number of runs. In our opinion, a quantification of this number does not add relevant information. In the revised version of the manuscript, we have done some rewording to explain this better. We also added relevant literature for reference, e.g. Tarquini & Favalli (2015). In contrast to the "Fogo scenario" hazard maps (Fig. 10), the "Pico Pequeno scenario" hazard maps (Fig. 9) contain areas of both, 100 % and 0 % probability of invasion. This is why we only introduce a lower threshold of probability in Fig. 10 (0 % probability only occurs on top of cones or at some places along the Bordeira wall in the "Fogo scenario"). In Fig. 9, no color is 0 probability of invasion (because there is a restricted area of vent opening in Fig. 9).

- Fig.8b: this is a key results of the study which could be better valorized. An histogram of the thickness distribution should be provided. The color bars suggest values from -12.8 to +52.7 m: what proportion of the thickness are negative, how could this be explained and how does this impact the total volume? Author mention a maximum thickness value of 35m: why does the color bar goes to 52 m then?

Author reply: We will provide a histogram of the thickness distribution. Negative values occur at the edges of the flow, where ground subsidence is observed due to loading (also obvious from InSAR data). Also, negative values occur close to the vent due to vent opening. The lava flow thickness is indeed maximal ∼35 m, but the cone at the eruption site built up to be maximal 52.7 m. We will add this to the text and provide an explaining figure.

- Section 4.2: the aeral coverage of the lava flow on lines 18 and 28 (page 9) are not matching. Please provide an histogram of the thickness values (at the moment you give

the mean and some max value, but you don't provide the distribution nor the minimum values that are probably below zero: Fig. 8b).

Author reply: The area we use is calculated from the coherence maps. We actually also used the DEM difference map to check the area and the value was slightly smaller (as we are missing data at the westernmost tip of the northern lava lobe). We have therefore deleted the area from Sect. 4.2. As mentioned above, we will provide a histogram of the thickness values in the final version of the manuscript.

- Page 11: the pre-post hazard map comparison is interesting at the caldera scale for the 'Pico Pequeno' scenario. It is less relevant for the 'Fogo scenario' as the changes are minor and similar to the ones observed in previous scenario. I advise to shorten or cut lines 11-18 (page 11).

Author reply: We agree and we will shorten the paragraph.

- Page 12, section 5.1: this section could be largely reduced as it brings little new information. The technique of coherence loss to map new volcanic products is indeed quite standard and does not disserve a long discussion. I disagree with the sentence (line 26-): "as the resulting extent of the mapped and the simulated 2014-15 lava flows match almost perfectly". Looking at fig. 8a it is obvious that this is correct only for specific zones andonly for the early emplacement stage of the lava flow.

Author reply: It is true that the good fit is only correct for specific zones and only for the early phase of the eruption. We will shorten and reword this paragraph in the final version of the manuscript accordingly. We will also mention here the comparison to Cappello et al. (2016), who achieved a better fit using a deterministic model.

- Page 15, lines 17-28: this paragraph does not relate at all to the presented results. Although I know that the modelling approach might enable to simulate the influence and optimal location of a barrier, this is not done in this case, and I doubt this would be practical solution for the Cha caldera, since the eruption probability and the settlements

are dispersed. The example of Chirico et al. (2009) mentioned is a good example of a purely theoretical modelling exercise with no applicability on the ground. I would advice to cut this paragraph.

Author reply: We agree and we will shorten the paragraph (not mentioning the barriers anymore).

———————————————————